
# 1 Synthesis of Global Actual Evapotranspiration from 1982 to 2019

Abdelrazek Elnashar[1,2,3], Linjiang Wang[1,2], Bingfang Wu[1,2*], Weiwei Zhu[1], Hongwei Zeng[1,2]
[1]State Key Laboratory of Remote Sensing Science, Aerospace Information Research Institute, Chinese Academy of
Sciences, Beijing, 100094, China
[2]College of Resources and Environment, University of Chinese Academy of Sciences, Beijing, 100049, China
[3]Department of Natural Resources, Faculty of African Postgraduate Studies, Cairo University, Giza, 12613, Egypt
*Correspondence to:* Bingfang Wu (wubf@aircas.ac.cn)
**Abstract.** As a linkage among water, energy, and carbon cycles, global actual evapotranspiration (ET) plays an
essential role in agriculture, water resource management, and climate change. Although it is difficult to estimate ET
over a large scale and for a long time, there are several global ET datasets available with varied algorithms, parameters,
and inputs, and they produce different levels of uncertainties. In this study, we propose a long-term synthesized ET
product at a kilometer spatial resolution and monthly temporal resolution from 1982 to 2019. Through a site-pixel
validation of certain global ET products over different land surface types and conditions, the high performing products
were selected through a high-quality flux eddy covariance covering the entire globe. According to the study results,
Penman-Monteith Leuning (PML), operational Simplified Surface Energy Balance (SSEBop), Moderate Resolution
Imaging Spectroradiometer (MODIS, MOD16A2105) and the Numerical Terradynamic Simulation Group (NTSG)
ET products were chosen to create the synthesized ET set. The proposed product agreed well with flux EC ET over
most of the all comparison levels, with a maximum ME (RME) of 13.94 mm (17.13 %) and a maximum RMSE
(RRMSE) of 38.61 mm (47.45 %). Furthermore, the product performed better than local ET products over China, the
United States, and the African continent and presented an ET estimation across all land cover classes. While no product
can perform best in all cases, the proposed ET can be used without looking at other datasets and performing further
assessments. Data are available on the Harvard Dataverse public repository through the following Digital Object
Identifier (DOI): https://doi.org/10.7910/DVN/ZGOUED (Elnashar et al., 2020) as well as it is available as Google
Earth Engine (GEE) application through this link: https://elnashar.users.earthengine.app/view/synthesizedet.
**1. Introduction**
Over most of the global land area, terrestrial evapotranspiration (ET) considers the second largest element of
the hydrological cycle after precipitation (Waring and Running, 2007b;Bastiaanssen et al., 2014) and represents the
linkage between water, energy, and carbon cycles (Gentine et al., 2019;Yang et al., 2016;Ferguson and Veizer, 2007)
and ecosystem services (Almusaed, 2011;Yang et al., 2015;Revelli and Porporato, 2018).
Hence, the accurate estimation of global ET is essential for understanding the global hydrological cycle and
water budgets (Oki and Kanae, 2006;Trenberth et al., 2007;Rodell et al., 2015), global drought (Sheffield et al.,
2012;Naumann et al., 2018;Spinoni et al., 2019;Lu et al., 2019;Forootan et al., 2019), impacts of climate change
(Waring and Running, 2007a;Zomer et al., 2008;Scheff and Frierson, 2014;Pan et al., 2015), climate change and global
water resources (Arnell, 1999;Haddeland et al., 2014;Arnell and Lloyd-Hughes, 2014), global transboundary basin



water scarcity (Degefu et al., 2018), water competition within a basin (Scott et al., 2001) and water stress/conflict
within transboundary basins (Samaranayake et al., 2016;Munia et al., 2016;Bastiaanssen et al., 2014).

While precipitation and runoff, which are other paramount factors of the global water balance, can be directly

measured by in situ weather stations and stream gauge networks as well as the availability of several datasets of
remotely sensed precipitation (Funk et al., 2015;Ashouri et al., 2015;Huffman et al., 1997;Yamamoto and Shige,
2015), it is difficult to measure ET, especially at large spatial scales (Senay et al., 2012;Zhang et al., 2016).

Recently, several global ET datasets have become available for a variety of purposes, and they have been

generated using remote sensing models, land surface models (LSM), and hydrological models (Trambauer et al.,
2014;Li et al., 2018;Sörensson and Ruscica, 2018). There are many differences among these models in relation to their
algorithms, parameters, and inputs, and they produce different levels of uncertainty (Wang and Dickinson, 2012;Xu
et al., 2019;Weerasinghe et al., 2019;Vinukollu et al., 2011a). The remote sensing model, which mainly focuses on
thermal remote sensing and the energy balance equation, will be represented by MOD16A2 (Mu et al., 2011), PML
(Zhang et al., 2019), SSEBop (Senay et al., 2013), SEBS (Chen et al., 2013), NTSG (Zhang et al., 2010), and GLEAM
v3.3b (Miralles et al., 2011b). The land surface model uses quantitative methods to simulate the vertical exchanges of
water and energy fluxes between the atmosphere and the land surface, as represented by GLDAS ET (Rodell et al.,
2004), GLEAM v3.3a (Miralles et al., 2011b), and FLDAS (McNally et al., 2017). TerraClimate, which is a
hydrological model, is based on a one-dimensional water balance approach (Abatzoglou et al., 2018). However, the
availability of many datasets introduces challenges related to how users choose the appropriate dataset for their
purposes (Wu et al., 2020).

Some studies have evaluated global ET products using an inferred estimate of ET obtained by subtracting

discharge (Q) from precipitation (P), ET = P - Q, over global river basins (Zhang et al., 2010;Vinukollu et al.,
2011a;Vinukollu et al., 2011b), congenital river basins (Weerasinghe et al., 2019), transboundary river basins (Hofste,
2014), and national river basins (Zhong et al., 2020). Some, on the other hand, have used the ensemble ET product as
observed data for evaluating certain ET products (Hofste, 2014;Trambauer et al., 2014;Andam-Akorful et al.,
2015;Bhattarai et al., 2019).

Site-pixel-level validation of certain ET products against flux EC ET as typically observed data has been

performed by several studies in specific regions (e.g., globally (Leuning et al., 2008;Zhang et al., 2010;Ershadi et al.,
2014;Michel et al., 2016); Asia (Kim et al., 2012); South Africa (Majozi et al., 2017); Europe (Ghilain et al., 2011;Hu
et al., 2015); North America (Jiménez et al., 2009;Mu et al., 2011); Europe and the United States (Miralles et al.,
2011b); the United States (Vinukollu et al., 2011b;Velpuri et al., 2013;Xu et al., 2019); and China (Jia et al., 2012;Liu
et al., 2013;Chen et al., 2014b;Tang et al., 2015;Yang et al., 2017;Li et al., 2018)).

Few previous studies have focused on merging certain ET products to create an ensemble ET product; for

instance, (Vinukollu et al., 2011a;Mueller et al., 2013;Badgley et al., 2015). They used all ET products and created a
merged product with a low spatial resolution. There are some global merged benchmarking evaporation products.
Vinukollu et al. (2011a) generated an ensemble of six global ET datasets at a daily time scale and 0.5°×0.5° (≈55 km)
spatial resolution for 1984-2007 using two surface radiation budget products and three models (i.e., surface energy
balance, revised Penman-Monteith, and modified Priestley-Taylor). They reported that the ensemble simple mean



value was reasonable; however, it was generally highly biased globally. Mueller et al. (2013) presented two monthly
global ET products that differed in their input ET members and temporal coverage. The first dataset consisted of 40
datasets for the period 1998-1995, while the second dataset merged 14 datasets from 1989 to 2005. Their ET was
derived from satellite and/or in situ observations (diagnostic) or calculated via LSM driven with observation-based
forcing or output from atmospheric reanalysis. Hence, they provided four merged synthesis products, one including
all datasets and three including datasets of each category (i.e., diagnostic, LSM, and reanalysis). They introduced the
first benchmark products for global ET and found that its multi-annual variations showed realistic responses and were
consistent with previous findings. Badgley et al. (2015) used a Priestly-Taylor Jet Propulsion Lab (PT-JPL) model
with 19 different combinations of forcing data to produce global ET estimates from 1984 to 2006 at a 1°×1° (≈100
km) spatial resolution. The ensemble ET members changed according to the number of products available each year,
which ranged between 4 and 12 members for 1999/2000 and 2001/2002, respectively. Their study focused on the
uncertainty in global ET estimates resulting from each class of input forcing datasets.
However, from the aforementioned studies, we can report three findings: (1) no single ET product performed
better than any other over different land surface types and conditions, (2) no one generated a single dataset for users,
and (3) the created ensemble ET products relied on several individual ET products and were not based on the product
with the best performance.
From our point of view, this work attempts to add to the growing scientific literature using a high-quality
dataset from global flux towers for further validations and inter-comparison between different global ET products to
understand their behavior within defined land cover types, elevation levels, and climatic classes. Moreover, we attempt
to build an ensemble ET product that has a minimum level of uncertainty over as many conditions as possible. The
study has two objectives: (1) to assess global ET products with in situ data derived from global flux towers across a
variety of land surface types and conditions to gain a better understanding of the disparities among datasets and (2) to
synthesize an ensemble global ET product with minimum uncertainties over more land surface types, climate systems,
and monthly, annually and interannual time steps for a longer time.

## 2. Data

### 2.1. Evapotranspiration

Twelve global ET datasets were explored in the current study (Table 1 and Appendix A). Of them, 5 datasets
used MODIS as input, including two versions (V6 and V105) of Moderate Resolution Imaging Spectroradiometer
(MODIS) Global Evapotranspiration (MOD16A2), Penman-Monteith Leuning ET product (PML), the operational
Simplified Surface Energy Balance ET (SSEBop) and the Surface Energy Balance System (SEBS). One dataset used
the Advanced Very High-Resolution Radiometer (AVHRR) as input, including the Numerical Terradynamic
Simulation Group (NTSG). The remainder mainly uses meteorological datasets as direct input, including field
measurements such as TerraClimate and reanalysis datasets such as FLADS and GLADS. The algorithm used in 12
global ET datasets is mainly the Penman-Monteith model, except for FLADS and GLDAS, which use the LSM, and
TerraClimate, which uses the soil water balance model. Priestley-Taylor is used to estimate evaporation from open





water by NTSG while Penman evapotranspiration is used in PML for water body, snow and ice evaporation. SSEBop,
SEBS, NTSG, and GLEAM are individually managed, and other ET products and elevation data are available from
GEE.
**Table 1.** Global ET products.

| Product | Method | Satellite data | Meteorological data | Resolution | | Temporal coverage |
|---|---|---|---|---|---|---|
| | | | | Spatial | Temporal | |
| MOD16A2 V6 | P-M, surface conductance | MODIS | GMAO | 500 m | 8 days | Jan 1, 2001 - Ongoing |
| MOD16A2 V105 | | | | 1 km | | Jan 1, 2000 - Dec 31, 2014 |
| PML | PML | | GLDAS V21 | 500 m | 8 days | Jul 4, 2002 - Dec 27, 2017 |
| SSEBop | P-M | | GDAS, PRISM | 1 km | 1 month | Jan 1, 2003 - Ongoing |
| SEBS | RS-SEB | MODIS, GLASS, GLAS | ERA-Interim | 5 km | 1 month | Jan 1, 2001 - Dec 31, 2010 |
| NTSG | Modified P-M & P-T | AVHRR | NCEP/NCAR Reanalysis | 8 km | 1 month | Jan 1, 1982 - Dec 31, 2013 |
| GLEAM 3.3b | P-T, soil stress factor | Radiation & air temperature | Certain reanalysis data | 0.25° | 1 month | Jan 1, 2003 - Dec 31, 2018 |
| GLEAM 3.3a | | - | | 0.25° | 1 month | Jan 1, 1980 - Dec 31, 2018 |
| FLADS | LSM | MODIS-IGBP, UMD-AVHRR | MERRA-2, CHIRPS | 0.10° | 1 month | Jan 1, 1982 - Dec 1, 2019 |
| GLDAS V20 | LSM | MCD12Q1, MOD44W, GTOPO30 | NOAA/GDAS, GPCP, AGRMET | 0.25° | 3 hours | Jan 1, 1948 - Dec 31, 2010 |
| GLDAS V21 | LSM | | | 0.25° | 3 hours | Jan 1, 2000 - Dec 23, 2019 |
| TerraClimate | SWB | Root zone storage capacity | WorldClim V1.4&2, CRU Ts4.0, JRA-55 | 0.25° | 1 month | Jan 1, 1958 - Dec 1, 2018 |

Note: P-M: Penman-Monteith; PML: P-M Leuning; P-T: Priestley-Taylor; RS-SEB: remotely sensed surface energy balance; LSM:
land surface model; SWB: soil water balance; GMAO: Global Modelling and Assimilation Office for daily meteorological
reanalysis data; GDAS: Global Data Assimilation System; PRISM: Parameter-elevation Regressions on Independent Slopes Model;
GLASS: Global Land Surface Satellite; GLAS: Geoscience Laser Altimeter System; MERRA-2: Modern-Era Retrospective
analysis for Research and Applications version 2; CHIRPS: Climate Hazards Group InfraRed Precipitation with Station data; RFE2:
The African Rainfall Estimation version 2.0; NOAA: National Oceanic and Atmospheric Administration; GPCP: Global
Precipitation Climatology Project; AGRMET: Agricultural Meteorological modelling system; CRU Ts4.0: Climate Research Unit
time series data version 4.0; JRA-55: Japanese 55-year Reanalysis.
Three regional ET datasets were used for comparison of consistent agreement over China, the United States
and the African continent (Table 2). Over China Mainland, The China Terrestrial ET product was used (Ma et al.,
2019); it is estimated monthly at a 0.1° (≈10 km) spatial resolution over 1982-2012 and can be retrieved from
http://en.tpedatabase.cn/. For the United States, daily SSEBop was used (Savoca et al., 2013;Senay and Kagone,
2019). These data are produced at a 0.009°×0.009° (≈1 km) grid cell spatial resolution from 2000 to 2018 and can be
downloaded from https://earlywarning.usgs.gov/ssebop/modis/daily. Daily SSEBop are aggregated to monthly time
step to be comparable with the synthesized ET temporal resolution. The Food and Agriculture Organization (FAO)
Water Productivity through Open access of Remotely sensed derived ET product (FAO WaPOR version 2) was used
for Africa (FAO, 2018, 2020). These data estimates are the sum of ET and interception, provided at a 0.002°×0.002°



(≈250 m) spatial resolution with a monthly temporal resolution from 2009; there are available from the following
website: https://wapor.apps.fao.org/home/WAPOR_2/1.
**Table 2.** Regional ET products.

| Product | Method | Satellite data | Meteorological data | Resolution | | Temporal coverage |
|---|---|---|---|---|---|---|
| | | | | Spatial | Temporal | |
| China Terrestrial ET | P-M, P-T | | CMFD | 10 km | 1 month | Jan 1, 1982 - Dec 31, 2012 |
| SSEBop | P-M | MODIS | NASA GDAS | 1 km | 1 day | Jan 1, 2012 - Dec 31, 2018 |
| WaPOR | RS-SEB | | MERRA/GEOS-5, CHIRPS | 250 m | 1 month | Jan 1, 2009 - Ongoing |

Note: P-M: Penman-Monteith; P-T: Priestley-Taylor; RS-SEB: remotely sensed surface energy balance; CMFD: China
Meteorological Forcing Dataset; NASA GDAS: National Oceanic and Atmospheric Administration's (NOAA) Global Data
Assimilation System; MERRA: Modern-Era Retrospective Analysis for Research and Applications; GEOS-5: Goddard Earth
Observing System, Version 5; CHIRPS: Climate Hazards Group InfraRed Precipitation with Stations.
**2.2. Flux EC data**
Comprehensive flux EC ET data from 645 sites (Fig. 1 and Table 3), AmeriFlux (ameriflux.lbl.gov);
FluxNET (fluxnet.fluxdata.org); EuroFlux (www.europe-fluxdata.eu); AsiaFlux (www.asiaflux.ne); and ChinaFlux
(www.chinaflux.org, partially provided by the Chinese Ecosystem Research Network), were collected and processed
to examine the performance of different estimated ET products. The downloaded EC data are half-hourly text-type
data, while the periods of flux EC ET ranged from 1 year (12 months) to 21 years (252 months) from 1994 to 2019.
The gap-filling technique was applied to the downloaded in situ EC data (Reichstein et al., 2005). Different EC flux
sites were spatially distributed on the heterogeneous underlying surface, corresponding to different land cover types
according to the International Geosphere-Biosphere Programme (IGBP) classification system, which is recorded in
each flux attribute data. The in situ measured ET (mm day$^{-1}$) can be obtained by the half-hourly average latent heat
flux (LE, W·m$^{-2}$s$^{-1}$) through Eq. (1), (Su, 2002):

$$ET = \frac{\overline{LE}}{\lambda} \times 3600 \times 24 \tag{1}$$

Where $\overline{LE}$ (W·m$^{-2}$s$^{-1}$) is the daily average of the half-hourly average latent heat flux, and $\lambda$ is the latent heat of
evaporation. $\lambda$ varies with air temperature in hydrologic or agricultural system modeling but only to a small extent
(Walter et al., 2001), and the value acts directly on the accuracy of the estimated in situ measured ET. Considering
that there are very limited impacts of the changes in air temperature on the estimated in situ measured ET (Henderson-
Sellers, 1984; Li et al., 2018), the constant value of 2.45 MJ kg$^{-1}$ is fixed in the calculation above (Walter et al., 2001).



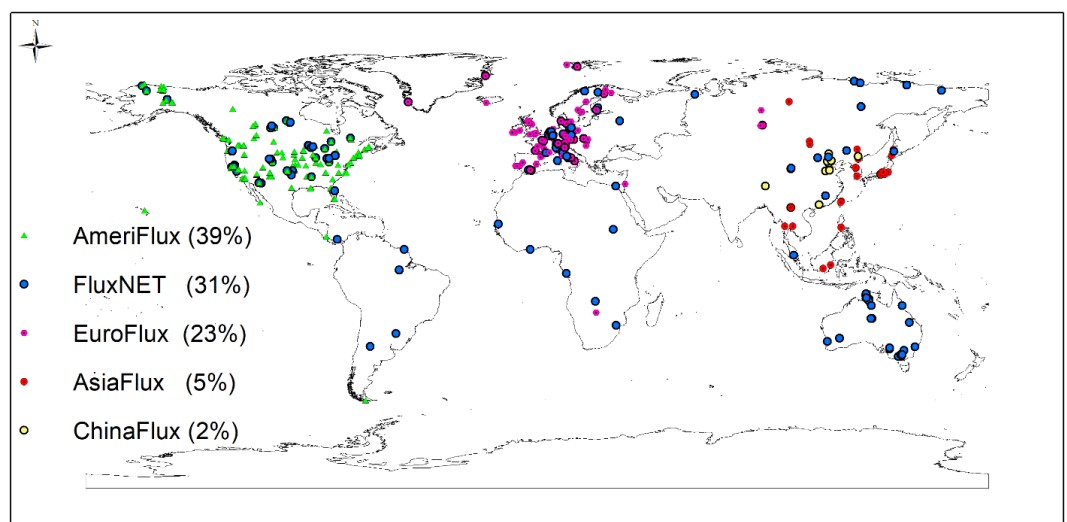

**Figure 1.** Spatial distribution of 645 in-situ flux EC sites across the world.

**Table 3.** Summary of 645 in-situ EC flux sites.

| Flux | Sites number | Time span | Elevation range (m) | Underlying surface IGBP type |
|---|---|---|---|---|
| AmeriFlux | 249 | 1994 to 2019 | -9 to 3199 | ENF/EBF/DBF/MF/CSH/OSH/WSA/SAV/GRA/WET/CRO/SNO/BSV/WAT |
| FluxNET | 203 | 1994 to 2019 | -10 to 4312 | ENF/EBF/DNF/DBF/MF/CSH/OSH/WSA/SAV/GRA/WET/CRO |
| EourFlux | 148 | 1996 to 2018 | -4 to 2436 | ENF/EBF/DBF/MF/CSH/OSH/WSA/SAV/GRA/WET/CRO/SNO |
| AsiaFlux | 33 | 2000 to 2015 | 0 to 3308 | ENF/EBF/DNF/DBF/MF/GRA/CRO/URB/WAT |
| ChinaFlux | 12 | 2003 to 2017 | 26 to 4317 | EBF/MF/GRA/CRO |

Note: ENF: Evergreen Needleleaf Forests; EBF: Evergreen Broadleaf Forests; DBF: Deciduous Broadleaf Forests; MF: Mixed Forests; CSH: Closed Shrublands; OSH: Open Shrublands; WSA: Woody Savannas; SAV: Savannas; GRA: Grasslands; WET: Permanent Wetlands; CRO; Croplands; URB: Urban and Build-up Lands; SNO: Permaneng Snow and Ice; BSV: Barren or Sparsely Vegetated Area; WAT: Water Bodies.

**2.3. Aridity index**

The mean global aridity index dataset was produced by (Zomer et al., 2008) using WorldClim global climate data. The aridity index was estimated as the mean annual precipitation divided by the mean annual potential evapotranspiration, and the latter was calculated by the Hargreaves equation. The spatial resolution was 0.0083°×0.0083° (≈1 km) grid cell (Trabucco and Zomer, 2018) and the data can be downloaded from https://cgiarcsi.community/data/global-aridity-and-pet-database.

**2.4. Elevation data**

The Shuttle Radar Topography Mission (SRTM) data were provided at a resolution of one arc-second and

void-filled (Farr et al., 2007). For the geographic areas outside the SRTM coverage area, the Global Multi-resolution
Terrain Elevation Data 2010 (GMTED2010), which have a resolution of 7.5 arc-seconds, were used (Danielson and
Gesch, 2011).
**3. Methods**
**3.1 Assessment**

Because ET is highly variable in both space and time (Schaffrath and Bernhofer, 2013;Fisher et al., 2017), a

comprehensive evaluation from different perspectives is required (Trambauer et al., 2014;McCabe et al., 2016;Li et
al., 2018). For each flux tower location, the aridity index, elevation and estimated ET data were extracted. The aridity
index was classified (Table 4), according to the United Nations Environment Programme definition (UNEP, 1997)
into four classes (i.e., humid: 361 (56 %), semiarid: 167 (26 %), dry sub-humid: 82 (13 %), and arid: 35 (5 %)).
Elevations were classified into three levels (i.e., <500 m: 452 (70 %), 500 m-1500 m: 135 (21 %), and >1500 m: 58
(9 %)). Land cover included five types (i.e., forests: 349 (54 %), grasslands: 128 (20 %), croplands: 89 (14 %), water
bodies: 73 (11%), and others (barren land and permanent snow and ice): 6 (1 %)). Accordingly, the following metrics
were estimated using Eqs. (2-7):

$$ME = \frac{1}{n}\sum_{i=1}^{n} Y_i - X_i \tag{2}$$

$$RME = \frac{ME}{X} \tag{3}$$

$$RMSE = \sqrt{\frac{\sum_{i=1}^{n}(Y_i - X_i)^2}{n}} \tag{4}$$

$$RRMSE = \frac{RMSE}{X} \tag{5}$$

$$R = \frac{\sum_{i=1}^{n}[(Y_i - Y)(X_i - X)]}{\sqrt{\sum_{i=1}^{n}(Y_i - Y)^2}\sqrt{\sum_{i=1}^{n}(X_i - X)^2}} \tag{6}$$

$$TS = \frac{4(1 + R)}{\left(std + \frac{1}{std}\right)^2 (1 + R_0)} \tag{7}$$

Where ME is the mean error; RME is the relative mean error; RMSE is the root mean square error; RRMSE is the
relative root mean square error; R is the correlation coefficient; TS is the Taylor score; n is the sample number; i is
the i$^{th}$ sample; X is the mean of the observed EC ET data; Y is the mean of different estimated ET data; std is the
standard deviation of the estimated ET normalized by the standard deviation of the observed EC ET; and $R_0$ is the
maximum theoretical R, with an $R_0$ value of 0.9976 (Taylor, 2001).

The magnitude of ME (the absolute value) is used as a bias indicator (Mu et al., 2011;Yang et al., 2017),

while its sign indicates whether different ET products overestimate or underestimate the flux EC ET values. The
accuracy of each ET product can be described by the RMSE (Miralles et al., 2011b;Hu et al., 2015). Moreover, the
relative values of RME and RRMSE are used for a fairer comparison between certain ET products among different





regions and periods (Majozi et al., 2017). In addition, correlation coefficients (R values) are used to measure the
strength of the relation between flux EC ET and different ET products (Ghilain et al., 2011;Hu et al., 2015), and with
the aid of the Taylor score (TS), the overall performance of each product can be described well (Taylor, 2001;Mu et
al., 2011). To rank each ET product, lower ME, RME, RMSE, and RRMSE values and higher R and TS values are
desired, lower biases and higher accuracies.
**Table 4.** Climate classification according to the global aridity index values.

| Aridity Index value | Climate class |
|---|---|
| <0.03 | Hyper arid |
| 0.03-0.20 | Arid |
| 0.20-0.50 | semiarid |
| 0.50-0.65 | Dry sub-humid |
| >0.65 | Humid |

**3.2 Synthesis method**
The current study proposes three steps to develop a synthesized global ET dataset. First, the ET datasets are
compared based on validated metrics, in which a matrix was developed to indicate level one and two validation metrics
of all ET products over all comparison levels, see Table 5. There are six validation criteria in rows (i.e., ME (mm),
RME (%), RMSE (mm), RRMSE (%), R, and TS) and 26 comparison levels in columns (i.e., monthly average (01),
annual average (02), monthly (January-December: 03-14), land cover types (15-19), climate classes (20-23), and
elevation levels (24-26)). The total number of cells is 156. Each cell represents a free competition between certain ET
products to occupy this cell based on each validation criterion. Then, selecting ET data for further synthesis, based on
the magnitudes (absolute values) of each validation index of all ET products across all comparison classes (01-26),
the best first and second levels of ET products within each cell were selected; additionally, the count and percent of
each ET product in all cells were calculated to calculate the total count and percent from levels one and two, see Table
6. All ET products will be sorted in descending order based on the total percentage of levels one and two. Finally, the
first two or three highly ranked ET products were incorporated into the ensemble ET. For that, the selected ET products
were resampled to a comparable spatial resolution if needed, and the average was used as the synthesized ET value.
**4. Results**
**4.1. Assessment of existing global ET datasets**
Figure 2 shows that seasonality exists and is captured well by all ET datasets, with some exceptions over
barren land, permanent snow and ice, and arid areas (not shown). The maximum ET occurs during July and differs
according to each ET dataset. Generally, MOD16A2 represents the minimum estimated ET across all conditions, while
the maximum ET is represented by SSEBop across all conditions except over humid regions and at elevations between
500 m and 1500 m. From Figures (3, 5-11), the best-fitted linear regression line (blue-solid line) compared to the 1:1
line (red-dashed line), all ET datasets overestimate the flux EC ET in lower ET values and underestimate the flux EC
ET in higher ET values with two exceptions. The first exception is over barren land and permanent snow and ice,
where MOD16A2 underestimates and GLDAS21, GLEAM33a, and TerraClimate overestimate under both lower and





higher ET values (not shown). Second, in dry sub-humid areas, GLDAS21 (Fig. 8e3) and SSEBop (Fig. 8c3)
overestimate under both lower and higher ET values. Applying the highest R (TS) and lowest error metrics role,
MOD16A2 cannot present any role; additionally, only one contribution by the lowest RRMSE was found in February
and the highest TS was found in March for TerraClimate and GLEAM33b, respectively.

### 4.1.2. Validation by all sites' monthly ET

Figure 3 shows that only SEBS and MOD16A2 underestimate flux EC ET. PML is the dataset that best agrees
with the observed ET, and it had the lowest RMSE (RRMSE). MOD16A2105 returned the smallest absolute ME,
while SEBS yielded the smallest RME. Figure 4 shows there are interannual differences between certain ET product
performances. MOD16A2 shows negative MEs and RMEs for all months, with larger biases during March, April, and
May, while FLDAS shows positive MEs and RMEs for all months, with larger biases during March, April May, June,
and July. For other products, the ME and RME signs vary among months; for instance, the ME and RME values of
GLDAS21 are negative (underestimated) during February, September, and November and positive (overestimated) in
the remaining months, with larger biases during March, April, May, June, and July. The RMSE declines from January
to February and then increases until July and declines again until November. The minimum RMSE values occur during
February, November, and December, while the maximum values occur during June, July, and August.
For instance, the RMSE in July ranges from 36.28 mm to 52.41 mm for FLDAS and PML, respectively,
while it ranges from 17.08 mm to 21.68 mm for PML and SEBS, respectively. RRMSE declines from January and
reaches its minimum in June and then increases again until December, except for SEBS in December. The highest
values of RRMSE (>80 %) occur in January, February, November, and December except for SEBS in December,
while the lowest values (<60 %) exist in June, July, and August. The R-value declines from January and reaches its
minimum in May; it then increases starting in August. Except for MOD16A2, all products have an R-value greater
than 0.60 during January, February, November, and December. SEBS has the lowest R-value during March, April,
May, and June, while PML yields the highest R-value during all months except January and December. Except for
MOD16A2 in February, which has a TS value above 0.60, as with the R-value, the TS declines from January and
reaches its minimum in May and then increases again starting in August. Figures 3 and 4 show these products yield
intra-annual ET variations but vary in their performance according to the selected validation metrics, which also vary
among all months (from January to December).



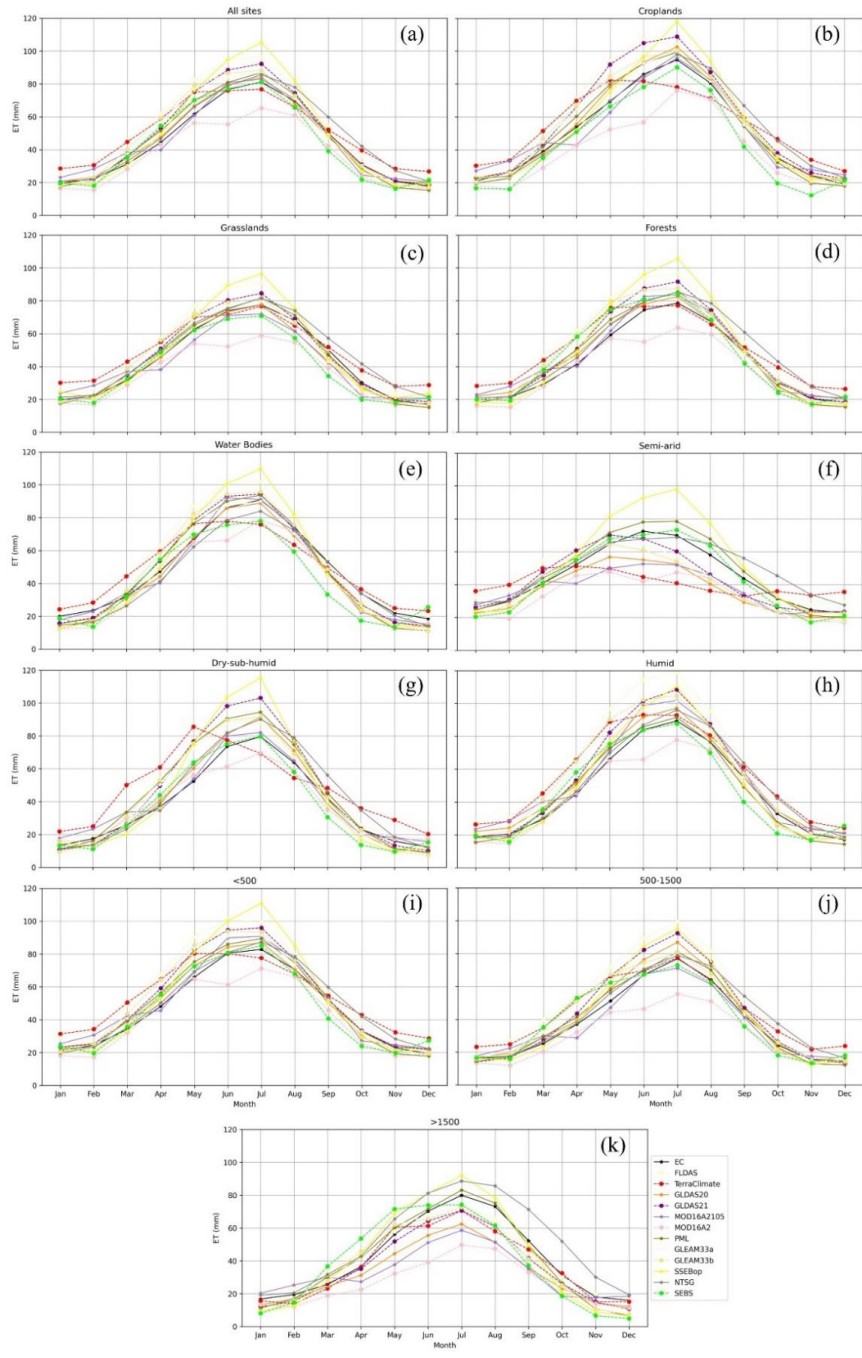

**Figure 2.** Monthly average ET products over all flux sites (**a**), land cover types (croplands: (**b**); grasslands: (**c**); forests: (**d**); water bodies: (**e**)), climate classes (semiarid: (**f**); dry sub-humid: (**g**); humid: (**h**)), and elevation levels (<500 m: (**l**), 500 m-1500 m: (**j**), and >1500m: (**k**)).

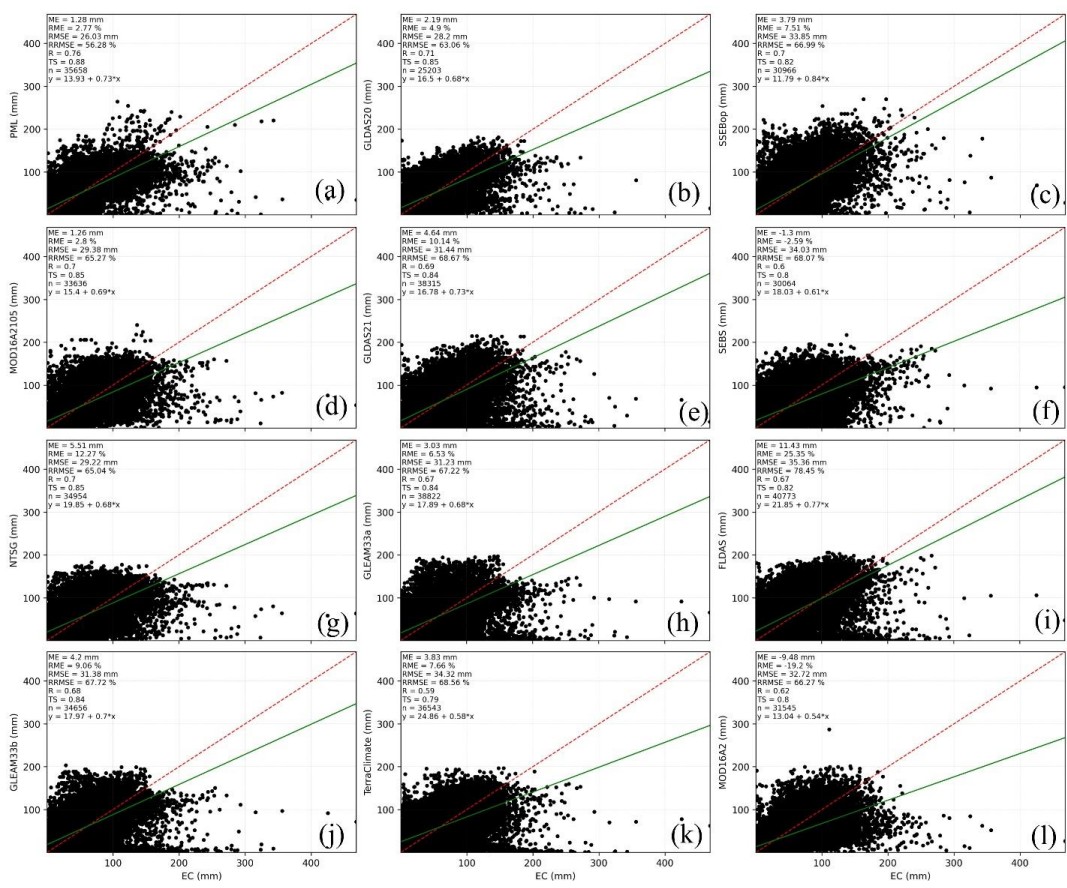

**Figure 3.** Monthly ET products (PML: (**a**); GLDAS20: (**b**); SSEBop: (**c**); MOD16A2105: (**d**); GLDAS21: (**e**); SEBS: (**f**); NTSG: (**g**); GLEAM33a: (**h**); FLDAS: (**i**); GLEAM33b: (**j**); TerraClimate: (**k**); MOD16A2: (**l**)) against flux EC ET aggregated for all sites.

Earth System
Science
Data

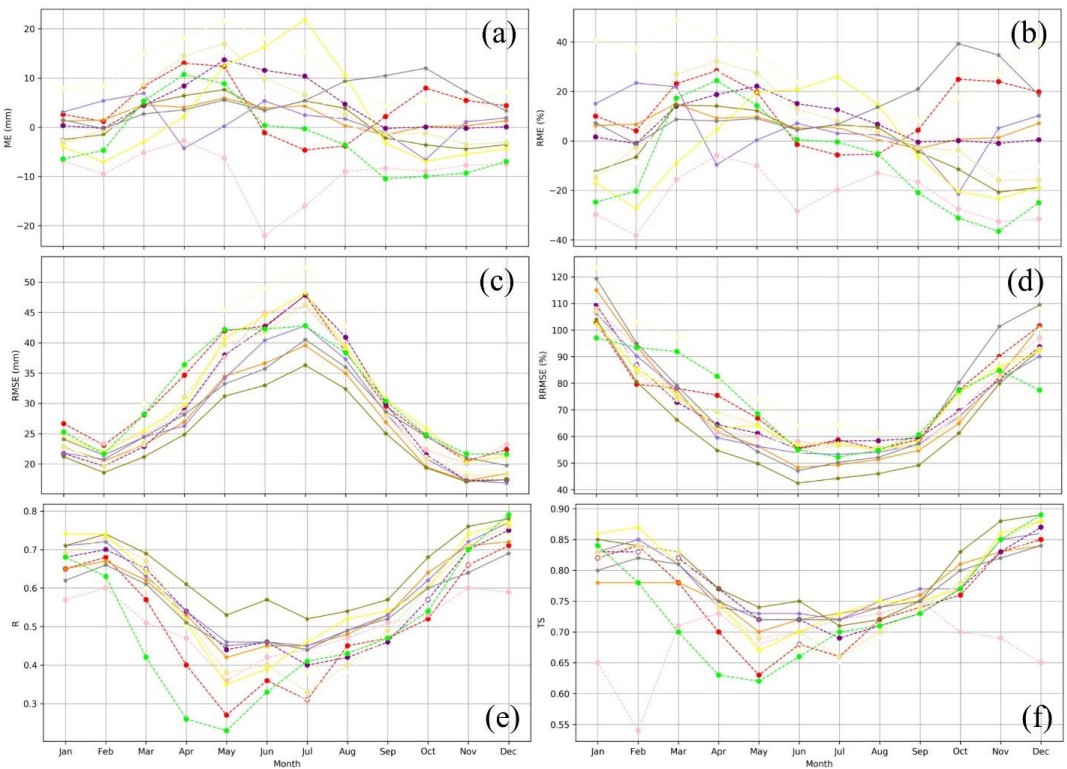

**Figure 4.** Monthly validation metrics (ME (mm): (**a**); RME (%): (**b**); RMSE (mm): (**d**); RRMSE (%): (**d**); R: (**e**); TS: (**f**)) of ET products against flux EC ET for all sites (legend as Figure 2k).

### 4.1.3. Validation by all sites' annual ET

Figure 5 shows all ET products overestimate the observed ET with two exceptions, SEBS and MOD16A2. In all environmental conditions, PML has the highest R (TS) and the lowest ME (RME) and RMSE (RRMSE). Figures 3 and 5 indicate the obvious error metrics of annual scale performances that are consistent with those that come from the monthly time step. The lowest and highest absolute values of ME (RME) for monthly ET exist in MOD16A2105 (SEBS) and FLDAS, respectively, while those for annual ET exist in PML and FLDAS, respectively. Furthermore, PML yields the largest R and TS values for monthly and annual ET, but the minimum values of R and TS were registered with TerraClimate and MOD16A2 for monthly and annual ET, respectively. This result may be attributed to the aggregation of monthly ET into annual values.



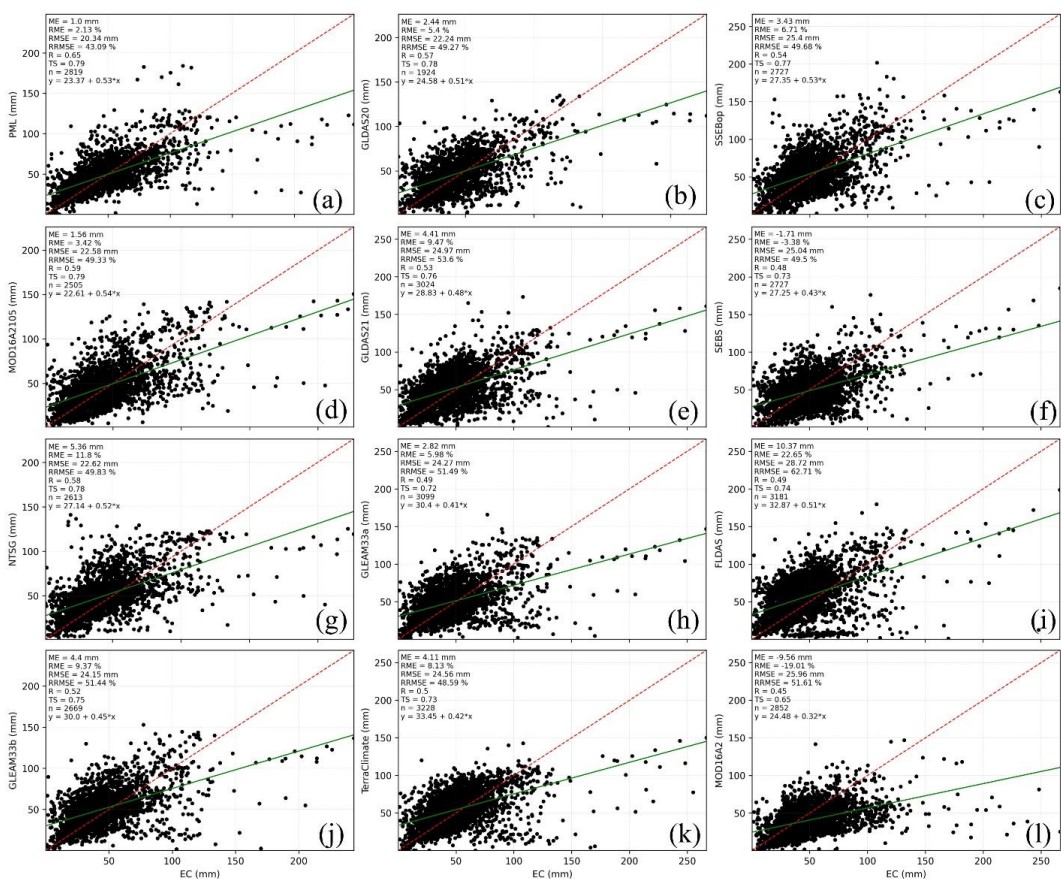

**Figure 5.** Annually ET products (PML: (**a**); GLDAS20: (**b**); SSEBop: (**c**); MOD16A2105: (**d**); GLDAS21: (**e**); SEBS: (**f**); NTSG: (**g**); GLEAM33a: (**h**); FLDAS: (**i**); GLEAM33b: (**j**); TerraClimate: (**k**); MOD16A2: (**l**)) against flux EC ET aggregated for all sites.

### 4.1.4. Validation by land cover types

Figures 6 and 7 show that, according to the ME (RME) sign, except for some ET products over croplands (i.e., MOD16A2, SEBS, MOD16A2105, and PML), grasslands (i.e., MOD16A2, SEBS, MOD16A2105, GLDAS20, and PML), forests (MOD16A2), and barren land and permanent snow and ice (i.e., MOD16A2105, MOD16A2, FLDAS, and GLDAS20), which underestimate the flux EC ET, the other ET products overestimate. For water bodies, MOD16A2105, GLEAM33b, GLDAS20, and FLDAS overestimate, while the other products produce underestimates. Over croplands, grasslands, and forests, PML is the best product for R (TS) and RMSE (RRMSE). Additionally, it has the highest TS over water bodies. The desired ME (RME) was obtained over croplands, grasslands, forests, water bodies, and barren land and permanent snow and ice by SSEBop, GLEAM33a, SEBS, NTSG, and GLDAS20, respectively. GLEAM33a also represents the highest R (TS) with the lowest RRMSE, while GLDAS20 has the smallest RMSE over barren land and permanent snow and ice. In addition, GLDAS20 has the lowest RMSE, while SSEBop has the highest R and lowest RRMSE over water bodies, see Table 5 (level one: 15-19).




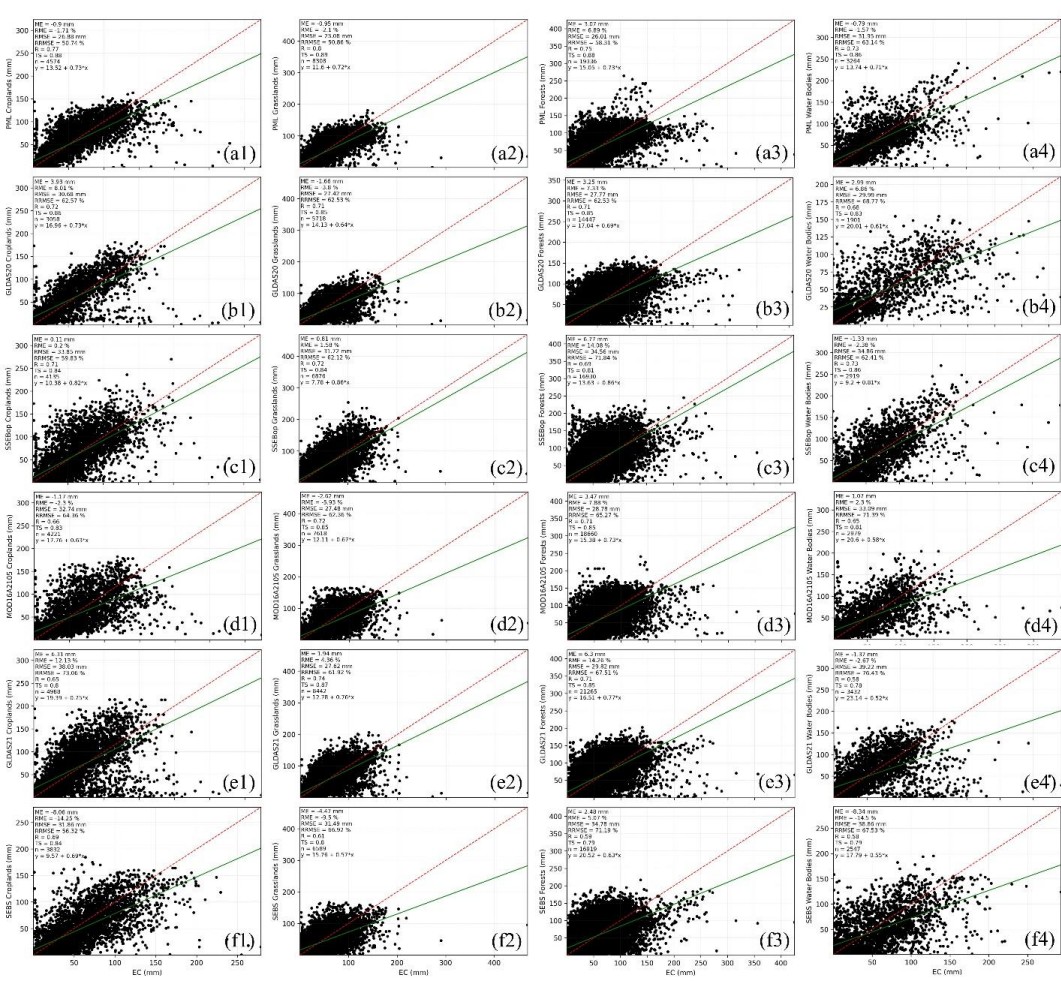

**Figure 6.** Monthly ET products (PML: (**a**); GLDAS20: (**b**); SSEBop: (**c**); MOD16A2105: (**d**); GLDAS21: (**e**); SEBS: (**f**)) against flux EC ET aggregated for all sites for each land cover type (croplands: (**1**); grasslands: (**2**); frosts: (**3**); water bodies: (**4**)).














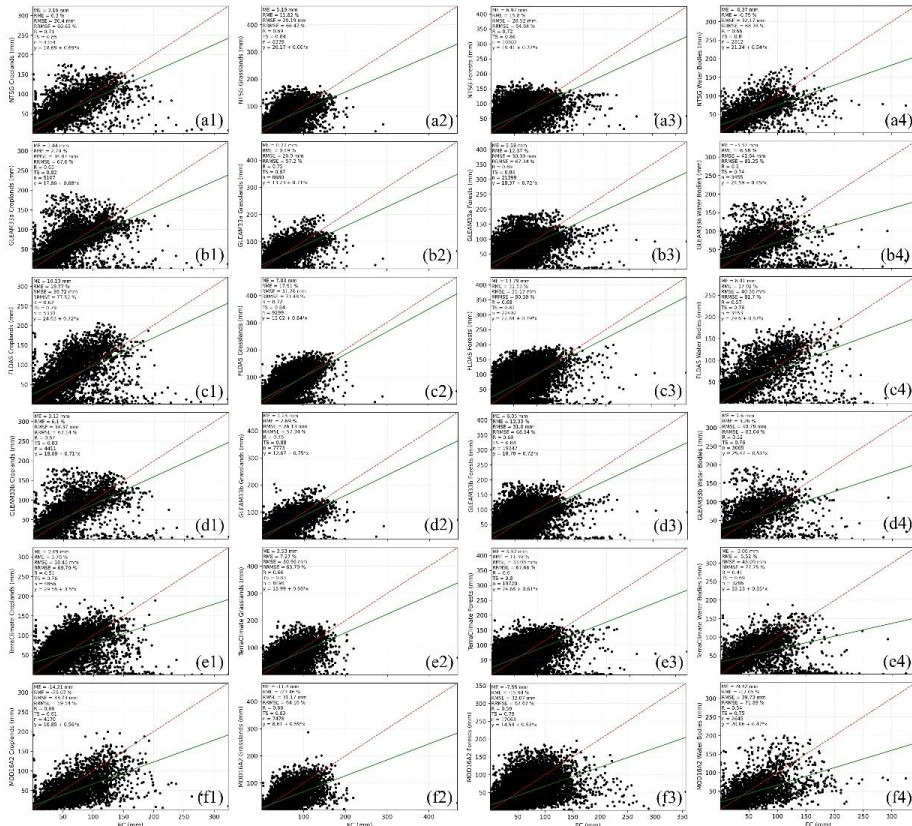

**Figure 7.** Monthly ET products (NTSG: (**a**); GLEAM33a: (**b**); FLDAS: (**c**); GLEAM33b: (**d**); TerraClimate: (**e**); MOD16A2: (**f**))
against flux EC ET aggregated for all sites for each land cover type (croplands: (**1**); grasslands: (**2**); frosts: (**3**); water bodies: (**4**)).
**4.1.5. Validation by climate classes**

Figures 8 and 9 show that SEBS, PML, NTSG, and SSEBop in arid areas and PML, NTSG, and SSEBop in

semiarid areas overestimate values, while MOD16A2 and SEBS in dry sub-humid areas and MOD16A2, SEBS, and
PML in humid areas underestimate values; for each aridity index class, other products were the opposite. Over humid
areas, PML represents the highest agreement and accurate dataset compared to the flux EC ET. Furthermore, it had
the highest R (TS) in the arid and semiarid areas and the smallest RMSE (RRMSE) in semiarid areas. GLDAS20
yielded the largest R (TS) with the smallest RMSE (RRMSE) in dry-sub-humid regions; over these regions,
MOD16A2105 presented the best ME (RME). FLDAS has two contributions, with the smallest ME (RME) and RMSE
(RRMSE) in semiarid and arid areas, respectively, while GLDAS21 has only one point over arid areas where the best
ME (RME) is found, see Table 5 (level one: 20-23).





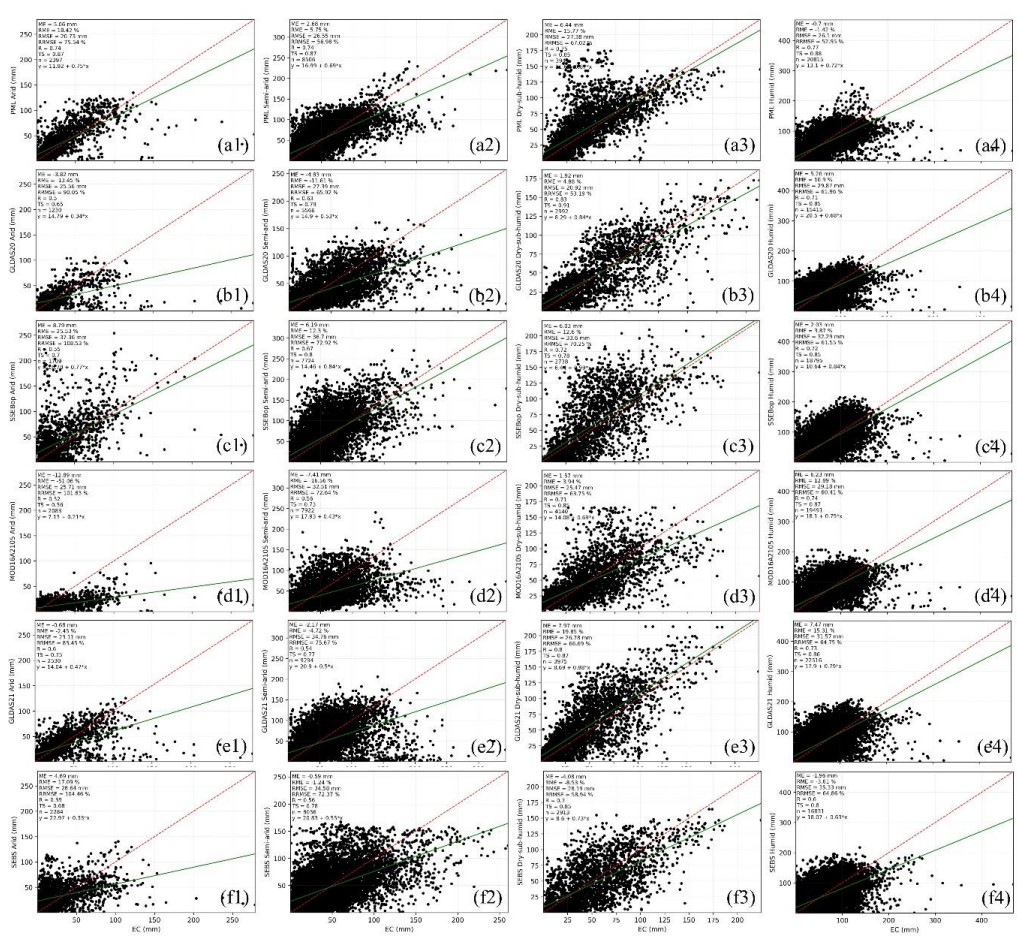

**Figure 8.** Monthly ET products (PML: (**a**); GLDAS20: (**b**); SSEBop: (**c**); MOD16A2105: (**d**); GLDAS21: (**e**); SEBS: (**f**)) against flux EC ET aggregated for all sites for each climate class (arid: (**1**); semiarid: (**2**); dry-sub-humid: (**3**); humid: (**4**)).



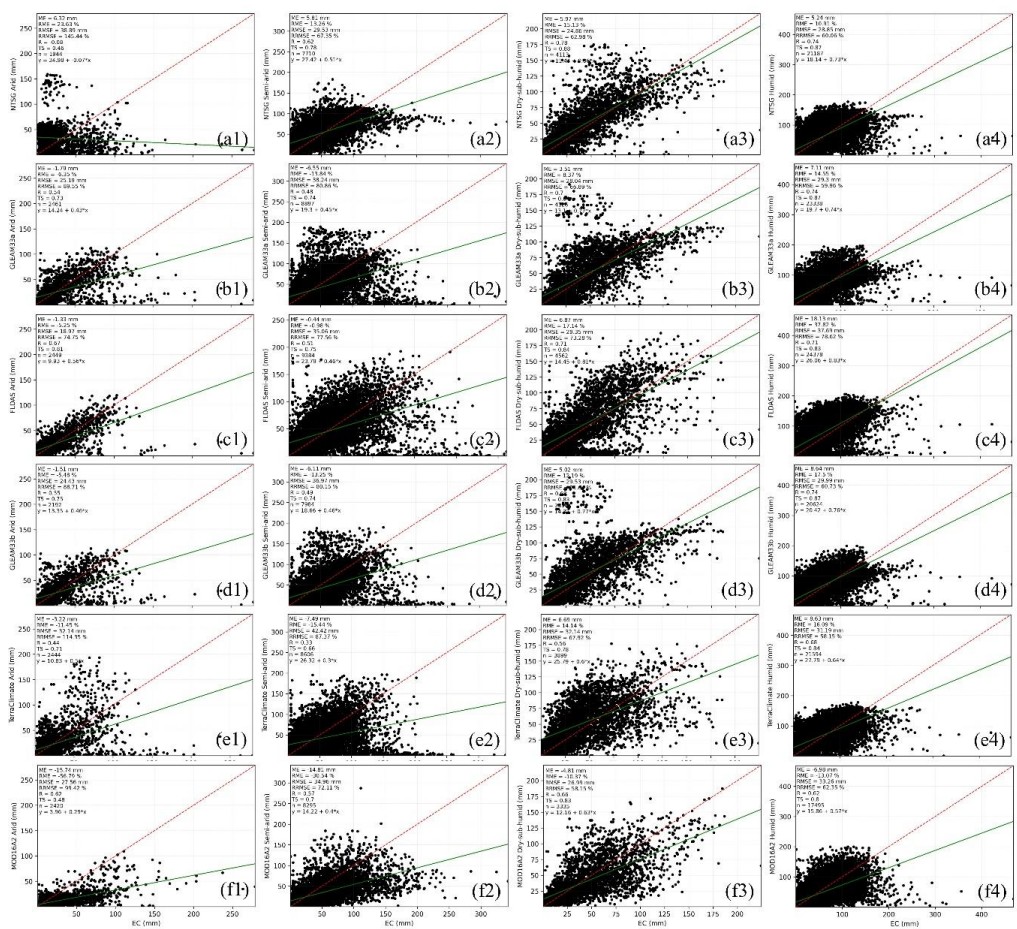

**Figure 9.** Monthly ET products (NTSG: (**a**); GLEAM33a: (**b**); FLDAS: (**c**); GLEAM33b: (**d**); TerraClimate: (**e**); MOD16A2: (**f**))
against flux EC ET aggregated for all sites for each climate class (arid: (**1**); semiarid: (**2**); dry-sub-humid: (**3**); humid: (**4**)).
**4.1.6. Validation by elevation levels**
Figures 10 and 11 show that MOD16A2 and SEBS over elevation levels <500 and MOD16A2 and
MOD16A2105 over elevation levels from 500 m to 1500 underestimate the values, while the other ET products
overestimate the values; additionally, at elevations >1500, only SSEBop and NTSG overestimate the values. The ET
product agreed best with the desired RMSE (RRMSE) in the PML product. Moreover, it yielded the best ME (RME)
at elevations <500 m. The preferred ME (RME) over elevations 500 m to 1500 m and elevations > 500 m was obtained
using SEBS and FLADS, respectively, see Table 5 (level one: 24-26).



**Figure 10.** Monthly ET products (PML: (**a**); GLDAS20: (**b**); SSEBop: (**c**); MOD16A2105: (**d**); GLDAS21: (**e**); SEBS: (**f**)) against flux EC ET aggregated for all sites for each elevation level (<500 m: (**1**); 500 m–1500 m: (**2**); >1500 m: (**3**)).





**Figure 11.** Monthly ET products (NTSG: (**a**); GLEAM33a: (**b**); FLDAS: (**c**); GLEAM33b: (**d**); TerraClimate: (**e**); MOD16A2:
(**f**)) against flux EC ET aggregated for all sites for each elevation level (<500 m: (**1**); 500 m–1500 m: (**2**); >1500 m: (**3**)).



### 4.2. Ensemble ET product

### 4.2.1. Ensemble steps

Table 5 provides the levels one and two validation metrics of all ET products for monthly (01), annual (02), interannual (January-December: 03-14), land cover types (croplands, grasslands, forests, water bodies, others: 15-19), climatic classes (arid, semiarid, dry sub-humid, humid: 20-23), and elevation levels (<500 m, 500 m-1500 m, >1500 m: 24-26). Each cell represents one of the validation levels (01-26) and the best-performing ET product based on the selected validation index, see Sect. 3.1.

Table 6 shows that, according to first-level accuracies, PML, GLDAS20, and SEBS represent the first three best-performing ET products, while according to the second level GLDAS20, PML, and MOD16A2105, and according to the total of the first and second levels, PML, GLDAS20, and SSEBop are the best, respectively. For example, PML yielded the best validation indices (the lowest ME, RME, RMSE, and RRMSE as well as the highest R and TS) over 83 (53 %) and 24 (15 %) cells in levels one and two, respectively; thus, the total count was 107 (34 %) cells. Accordingly, the three best-performing ET products over most of the all conditions are MPL followed by GLDAS20 (first level: 10 (6 %); second level: 37 (24 %); total: 37 (15 %)) and SSEBop (first level: 12 (8 %); second level: 15 (10 %); total: 27 (9 %)).

Since the three best-performing ET products differ in their spatial resolution and algorithms, we introduced an ensemble mean product at a 1000 m × 1000 m spatial resolution that spans from 2003 to 2017 (15 years) and relies on remotely sensed models (PML and SSEBop). It should be noted that although SEBS has one point more than SSEBop in the first level, it has 7 fewer points than SSEBop in the second level (5 %). In addition, SSEBop has a higher spatial resolution than that of SEBS. In the same manner, SSEBop and MOD16A2105 have the same performance in terms of total count (27 (9 %)), but SSEBop is higher by 5 points in the first level.

Obviously, from Table 7, the ensemble ET products cannot perform highly across all regions, and it had a total count of 50 %, followed by PML (44 %). Looking to the ensemble mean from Table 7 compared to PML from Table 6, the total count increased from 34 % to 50 % (+16 %), indicating that the ensemble mean, which created from PML and SSEBop, enhanced PML performance across all conditions by 16 % and PML itself still has the best performance by 44 %.

To introduce an ensemble product before 2003, first PML and SSEBop were ignored, and the same steps were repeated. Table 8 shows that the best-performing products are GLDAS20, MOD16A2105, and NTSG in terms of the total count. Since the last two products are based on remote sensing, they were selected to create the ensemble product before 2003 at a 1000 m × 1000 m spatial resolution. Although GLDAS20 agreed well over 42 % and had the lowest maximum ME among all datasets (9.73 mm), NTSG was selected to provide the ET estimates before 2000 because it had a higher spatial resolution, so it could capture more spatial details than GLDAS20.

Table 9 shows that the ensemble ET for 2001 and 2002 performed better than the original ET products, with values of 62 %, 38 %, and 50 % for level one, level two and the total, respectively. For the periods before 2001, NTSG can be used from 1982 to 2001 or GLDAS20 can be used instead. Hence, remotely sensed-based long-term ensemble ET can be synthesized from PML and SSEBop between 2003 and 2017 and from MOD16A2105 and NTSG between 2001 and 2002. SSEBop can be used after 2018, while before 2000, NTSG can be used.

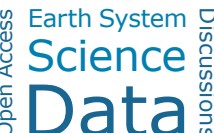

**Table 5.** Levels one and two validation metrics of all ET products for monthly (**01**), annually (**02**) interannual (January-December: **03**-**14**), land cover types (croplands, grasslands, forests, water bodies, others: **15**-**19**), climatic classes (arid, semiarid, dry sub-humid, humid: **20**-**23**), and elevation levels (<500 m, 500 m-1500 m, >1500 m: **24**-**26**), cells colour as Table **6**.

| Level | Indicator | 01 | 02 | 03 | 04 | 05 | 06 | 07 | 08 | 09 | 10 | 11 | 12 | 13 | 14 | 15 | 16 | 17 | 18 | 19 | 20 | 21 | 22 | 23 | 24 | 25 | 26 |
|---|---|---|---|---|---|---|---|---|---|---|---|---|---|---|---|---|---|---|---|---|---|---|---|---|---|---|---|
| Level one | ME | | | | | | | | | | | | | | | | | | | | | | | | | | |
| | RME | | | | | | | | | | | | | | | | | | | | | | | | | | |
| | RMSE | | | | | | | | | | | | | | | | | | | | | | | | | | |
| | RRMSE | | | | | | | | | | | | | | | | | | | | | | | | | | |
| | R | | | | | | | | | | | | | | | | | | | | | | | | | | |
| | TS | | | | | | | | | | | | | | | | | | | | | | | | | | |
| Level two | ME | | | | | | | | | | | | | | | | | | | | | | | | | | |
| | RME | | | | | | | | | | | | | | | | | | | | | | | | | | |
| | RMSE | | | | | | | | | | | | | | | | | | | | | | | | | | |
| | RRMSE | | | | | | | | | | | | | | | | | | | | | | | | | | |
| | R | | | | | | | | | | | | | | | | | | | | | | | | | | |
| | TS | | | | | | | | | | | | | | | | | | | | | | | | | | |

**Table 6.** The count, percent and the total count and percent of levels one and two of all ET products performance.

| Dataset | Level 1 count | Level 1 count (%) | Level 2 count | Level 2 count (%) | Total count | Total count (%) |
|---|---|---|---|---|---|---|
| PML | 83 | 53 | 24 | 15 | 107 | 34 |
| GLDAS20 | 10 | 6 | 37 | 24 | 47 | 15 |
| SSEBop | 12 | 8 | 15 | 10 | 27 | 9 |
| MOD16A2105 | 7 | 4 | 20 | 13 | 27 | 9 |
| GLDAS21 | 14 | 9 | 11 | 7 | 25 | 8 |
| SEBS | 13 | 8 | 8 | 5 | 21 | 7 |
| NTSG | 4 | 3 | 16 | 10 | 20 | 6 |
| GLEAM33a | 5 | 3 | 6 | 4 | 11 | 4 |
| FLDAS | 6 | 4 | 4 | 3 | 10 | 3 |
| GLEAM33b | 1 | 1 | 6 | 4 | 7 | 2 |
| TerraClimate | 1 | 1 | 6 | 4 | 7 | 2 |
| MOD16A2 | 0 | 0 | 3 | 2 | 3 | 1 |

**Table 7.** The count, percent and the total count and percent of levels one and two of PML and SSEBop products and their ensemble mean for the period 2003-2017.

| Dataset | Level 1 count | Level 1 count (%) | Level 2 count | Level 2 count (%) | Total count | Total count (%) |
|---|---|---|---|---|---|---|
| Mean | 43 | 28 | 113 | 72 | 156 | 50 |
| PML | 103 | 66 | 33 | 21 | 136 | 44 |
| SSEBop | 10 | 6 | 10 | 6 | 20 | 6 |

**Table 8.** The count, percent and the total count and percent of levels one and two of all ET products performance except PML and SSEBop.

| Dataset | Level 1 count | Level 1 count (%) | Level 2 count | Level 2 count (%) | Total count | Total count (%) |
|---|---|---|---|---|---|---|
| GLDAS20 | 42 | 27 | 27 | 17 | 69 | 22 |
| MOD16A2105 | 28 | 18 | 28 | 18 | 56 | 18 |
| NTSG | 14 | 9 | 35 | 22 | 49 | 16 |
| GLDAS21 | 23 | 15 | 14 | 9 | 37 | 12 |
| SEBS | 21 | 13 | 7 | 4 | 28 | 9 |
| GLEAM33a | 8 | 5 | 16 | 10 | 24 | 8 |
| GLEAM33b | 6 | 4 | 15 | 10 | 21 | 7 |
| FLDAS | 9 | 6 | 5 | 3 | 14 | 4 |
| TerraClimate | 3 | 2 | 5 | 3 | 8 | 3 |
| MOD16A2 | 2 | 1 | 4 | 3 | 6 | 2 |

**Table 9.** The count, percent and the total count and percent of levels one and two of NTSG and MOD16A2105 products and their ensemble mean for 2001 and 2002.

| Dataset | Level 1 count | Level 1 count (%) | Level 2 count | Level 2 count (%) | Total count | Total count (%) |
|---|---|---|---|---|---|---|
| Mean | 96 | 62 | 59 | 38 | 155 | 50 |
| NTSG | 19 | 12 | 68 | 44 | 87 | 28 |
| MOD16A2105 | 41 | 26 | 29 | 19 | 70 | 22 |

### 4.2.2 Contribution of ET datasets to the synthesized ET

The synthesized ET dataset was created at a 1000 m × 1000 m spatial resolution from 1982 to 2019 based on remotely sensed ET products. PML, SSEBop, MOD16A2105, and NTSG were augmented together to create the new dataset. Since SSEBop and MOD16A2105 have a 1000 m × 1000 m spatial resolution, PML was upscaled and NTSG was downscaled by pixel average and nearest neighbor resampling techniques in GEE, respectively. The synthesized ET was fully contributed by SSEBop for the years 2018 and 2019 and by NTSG from 1982 to 2000, while for the years 2001 and 2002, it was contributed by the simple mean of MOD16A2105 and NTSG. Finally, between 2003 and 2017, the value represents the simple mean of PML and SSEBop.

Since the synthesized ET performance was governed by each ET product(s) for the corresponding year from 1994 to 2019 (25 years), where the ET EC fluxes were available, most of the performance comes from PML and SSEBop for the 15 years from 2003 to 2017 (60 %), from MOD16A2105 and NTSG for 2 years (2001 and 2002; 8 %), from SSEBop for individual values in years 2018 and 2019 (8 %), and from NTSG for 7 years (24 %) from 1994 to 2000.

### 4.2.3. Synthesized global ET product

Figure 12 shows that, looking to July, except over barren land, permanent snow and ice, and arid areas (not shown), the maximum value of the synthesized ET lies between SSEBop, which yields the largest ET during all months, and PML. Hence, the long-term monthly synthesized ET is affected by PML and SSEBop more than by NTSG and MOD16A2105, as mentioned in Sect. 4.2.2.





Table 10 provides the average decadal synthesized ET (mm decade$^{-1}$) of the monthly, land cover types and
aridity index classes for all flux sites from 1994 to 2019. July represents the maximum synthesized ET, except during
2000-2009 (see Figs. 2 and 12). Across land cover types, croplands are higher than forests, followed by grassland,
where the average synthesized ET was 595, 548, and 539 for croplands, forests, and grasslands, respectively. Low
synthesized ET values across arid areas (average = 387 mm yr$^{-1}$) can be attributed to low vegetation cover. It should
be noted that Table 12 does not represent the perfect calculation of ET over each classification level because the total
number of fluxes for each class was not distributed well; for instance, in the arid areas, there were 35 (5 %) fluxes,
while in the humid area, there were 361 (56 %) fluxes.
Figure 13 shows the decadal (1982-1989, 1990-1999, 2000-2009, and 2010-2019) and long-term (1982-
2019) synthesized ET maps worldwide, except for Antarctica. Regarding the spatial distribution, the higher
synthesized ET was clustered in Malaysia, Singapore, and Indonesia and in the northern part of South America. During
the first and second decades, the synthesized ET was based on the NTSG product; thus, the same spatial distribution
was observed. Although the synthesized ET is mainly contributed by PML and SSEBop between 2003 and 2017, there
is little difference in their spatial distributions, where more ET can be observed during 2010-2019 over the northern
parts of South America.
Table 11 shows some statistics of the maps provided in Fig. 13 for all continents except Antarctica. The
standard deviation shows the ET variability across each continent; specifically, it is higher over Africa followed by
South America during the period of 2010-2019 and in Oceania. South America was followed by Africa, and Antarctica
returned the highest mean values of the synthesized ET, while the highest total ET came from Asia, South America,
and Africa, where the ET was 29.1 %, 21.7 %, 19.9 %, 16.7 %, 7.9 %, 4.2 %, and 0.5 % for Asia, South America,
Africa, North America, Europe, Australia, and Oceania, respectively. The maximum values of the synthesized ET
were recorded over Asia during the third decade followed by the fourth decade across all continents.
**4.2.4 Validation of the synthesized ET**
Figures 14-17 show that the synthesized ET agreed well with the observed data, where the $R^2$ (TS) ranged
between 0.70 (0.85) and 0.78 (0.89), except at the annual time step (Fig. 14b) and over barren land and permanent
snow and ice (not shown), where $R^2$ (TS) was 0.65 (0.81) and 0.68 (0.80), respectively. Based on the ME sign, the
value was underestimated only over water bodies. The magnitude of ME (RME) ranged between 0.54 mm (1.05 %)
and 6.76 mm (16.62 %), while the RMSE (RRMSE) ranged from 20.95 mm (45.22 %) to 30.12 mm (59.61 %).
Looking at the regression line equation, with no exceptions, the synthesized ET overestimated the flux EC ET at lower
ET values and underestimated the flux EC ET at higher ET values. As mentioned above, even the long-term
synthesized ET cannot perform best across all comparison levels (Tables 12 and 13).
During the periods 2018-2019 and before 2001, the synthesized ET performance came from the original
datasets of SSEBop and NTSG, respectively. The ensemble mean has a total count of 50 % over the periods 2003-
2017 and 2001-2002 compared to the original datasets, indicating that it can perform better than other ET products
over half of all comparison levels, see Tables 7 and 9.



**Figure 12.** Monthly average synthesized ET and the original products over all flux sites (**a**), land cover types (croplands: (**b**); grasslands: (**c**); forests: (**d**); water bodies: (**e**)), climate classes (semiarid: (**f**); dry sub-humid: (**g**); humid: (**h**)), and elevation levels (<500 m: (**l**), 500 m-1500 m: (**j**), and >1500m: (**k**)).




**Table 10.** The average decadal synthesized ET (mm decade$^{-1}$) of monthly, land cover types, and aridity index classes over all flux sites.

| Level | 1982-1989 | 1990-1999 | 2000-2009 | 2010-2019 | 1982-2019 |
|---|---|---|---|---|---|
| January | 222 | 201 | 173 | 232 | 207 |
| February | 241 | 237 | 258 | 203 | 235 |
| March | 394 | 365 | 382 | 363 | 376 |
| April | 506 | 581 | 524 | 573 | 546 |
| May | 856 | 734 | 867 | 771 | 807 |
| June | 910 | 1054 | 1069 | 885 | 980 |
| July | 1182 | 1140 | 984 | 1006 | 1078 |
| August | 938 | 958 | 920 | 892 | 927 |
| September | 672 | 604 | 581 | 657 | 629 |
| October | 343 | 310 | 472 | 470 | 399 |
| November | 287 | 215 | 293 | 190 | 247 |
| December | 229 | 227 | 184 | 163 | 201 |
| Croplands | 597 | 619 | 595 | 577 | 597 |
| Grasslands | 526 | 546 | 539 | 557 | 543 |
| Forests | 541 | 561 | 544 | 546 | 549 |
| Water Bodies | 499 | 517 | 519 | 534 | 518 |
| Others | 280 | 288 | 230 | 195 | 248 |
| Arid | 400 | 405 | 366 | 398 | 392 |
| Semiarid | 519 | 538 | 528 | 541 | 532 |
| Dry sub-humid | 479 | 498 | 498 | 511 | 504 |
| Humid | 577 | 600 | 582 | 577 | 583 |
| Elevation <500m | 551 | 570 | 570 | 579 | 568 |
| Elevation 500 m-1500 m | 498 | 519 | 484 | 484 | 496 |
| Elevation >1500 m | 557 | 583 | 506 | 471 | 527 |

Open Access  Earth System  Discussions
Science
Data

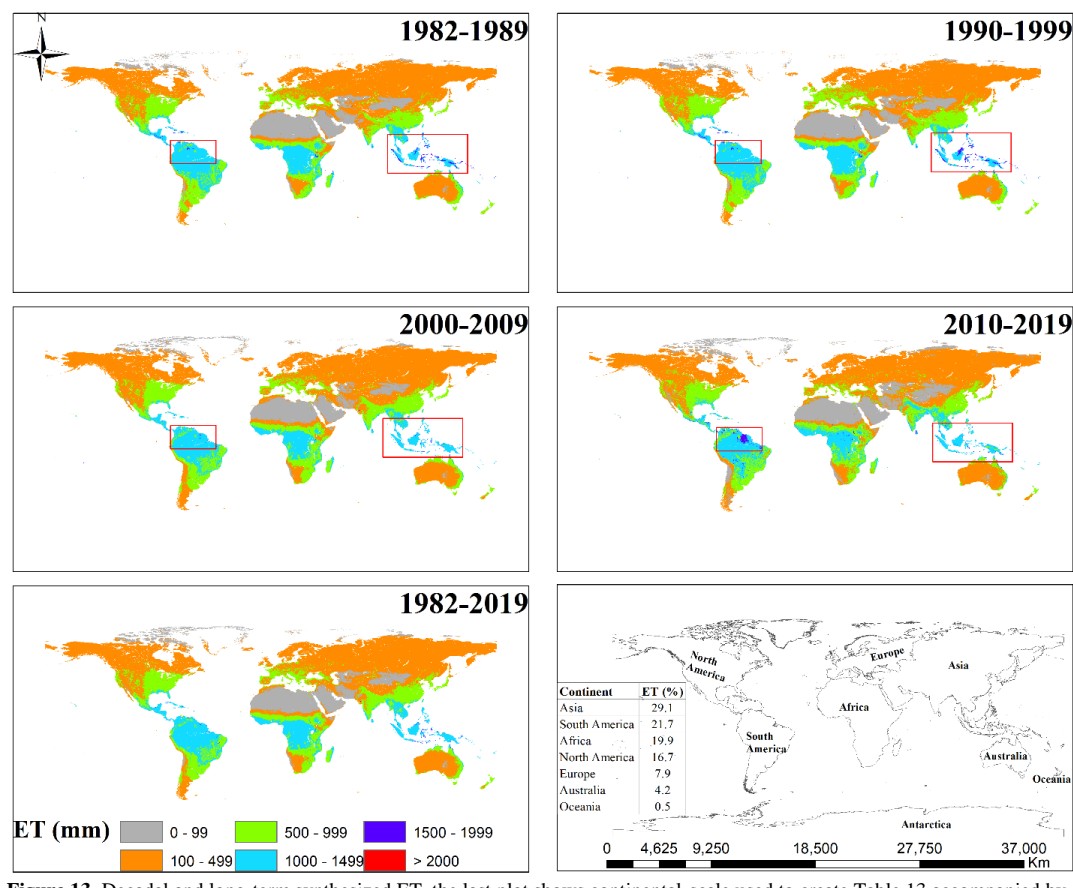

**Figure 13.** Decadal and long-term synthesized ET, the last plot shows continental-scale used to create Table 13 accompanied by the percent of ET over each continent for the periods 1982-2019 except Antarctica. Open the following link of GEE application to preview these maps: **https://elnashar.users.earthengine.app/view/synthesizedet**






**Table 11.** The statistics of the decadal and long-term synthesized ET (mm) extracted from the maps in Figure 13.

| Period | Continent | Min | Max | Mean | STD | Sum |
|--------|-----------|-----|-----|------|-----|-----|
| | Africa | 1 | 2367 | 536 | 491 | 16811613860 |
| | Asia | 6 | 2429 | 369 | 334 | 24117740643 |
| | Australia | 20 | 2641 | 443 | 232 | 3784646648 |
| 1982-1989 | Europe | 42 | 1534 | 400 | 152 | 6654513857 |
| | North America | 20 | 2436 | 409 | 285 | 14008259210 |
| | Oceania | 283 | 1805 | 882 | 370 | 396437575 |
| | South America | 42 | 2031 | 1003 | 329 | 18827424956 |
| | Africa | 0 | 2551 | 550 | 498 | 17261577174 |
| | Asia | 9 | 2478 | 378 | 337 | 24750672539 |
| | Australia | 20 | 2655 | 436 | 234 | 3720804592 |
| 1990-1999 | Europe | 42 | 1556 | 421 | 165 | 7001805714 |
| | North America | 19 | 2376 | 419 | 290 | 14381002938 |
| | Oceania | 267 | 1833 | 871 | 372 | 391403653 |
| | South America | 44 | 2055 | 1016 | 328 | 19076452222 |
| | Africa | 0 | 2323 | 537 | 459 | 16882958857 |
| | Asia | 0 | 3428 | 382 | 322 | 25221521620 |
| | Australia | 19 | 2129 | 417 | 215 | 3575447017 |
| 2000-2009 | Europe | 0 | 1370 | 406 | 131 | 6906585842 |
| | North America | 0 | 2369 | 383 | 269 | 14091347136 |
| | Oceania | 231 | 2173 | 805 | 364 | 386791335 |
| | South America | 12 | 2258 | 977 | 346 | 18432272851 |
| | Africa | 0 | 2845 | 540 | 477 | 17004524752 |
| | Asia | 0 | 2727 | 378 | 348 | 24953998863 |
| | Australia | 9 | 2840 | 375 | 245 | 3219071931 |
| 2010-2019 | Europe | 4 | 1557 | 379 | 125 | 6440972476 |
| | North America | 0 | 2829 | 391 | 280 | 14398921430 |
| | Oceania | 82 | 2501 | 728 | 387 | 349965725 |
| | South America | 0 | 2963 | 936 | 414 | 17665154856 |
| | Africa | 1 | 2357 | 541 | 473 | 17011325017 |
| | Asia | 1 | 2371 | 377 | 329 | 24895769068 |
| | Australia | 20 | 2326 | 419 | 220 | 3592380723 |
| 1982-2019 | Europe | 5 | 1440 | 400 | 137 | 6793072099 |
| | North America | 0 | 2403 | 390 | 277 | 14364800491 |
| | Oceania | 194 | 2206 | 825 | 369 | 396700784 |
| | South America | 8 | 2566 | 982 | 343 | 18533893930 |

Note: Min: Minimum; Max: Maximum; STD: Standard Deviation.






**Table 12.** Levels one and two validation metrics of all ET products (except MOD16A2) and the synthesized ET for monthly (**01**), annually (**02**) interannual (January-December: **03-14**), land cover types (croplands, grasslands, forests, water Bodies, others: **15-19**), climatic classes (arid, semiarid, dry sub-humid, humid: **20-23**), and elevation levels (<500 m, 500 m–1500 m, >1500 m: **24-26**), cells colour as Table **13**.

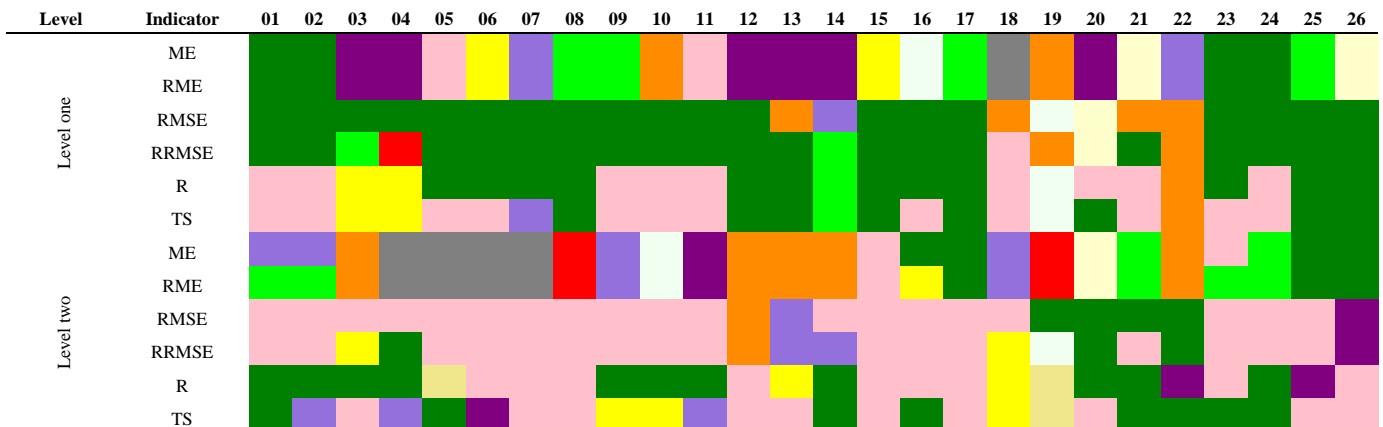


**Table 13.** The count, percent and the total count and percent of levels one and two of all ET products (except MOD16A2) and the synthesized ET performance.

| Dataset | Level 1 count | Level 1 count (%) | Level 2 count | Level 2 count (%) | Total count | Total count (%) |
|---|---|---|---|---|---|---|
| PML | 66 | 42 | 33 | 21 | 99 | 32 |
| Synthesized | 26 | 17 | 57 | 37 | 83 | 27 |
| GLDAS20 | 12 | 8 | 12 | 8 | 24 | 8 |
| GLDAS21 | 12 | 8 | 7 | 4 | 19 | 6 |
| SEBS | 12 | 8 | 7 | 4 | 19 | 6 |
| MOD16A2105 | 6 | 4 | 12 | 8 | 18 | 6 |
| SSEBop | 8 | 5 | 8 | 5 | 16 | 5 |
| NTSG | 2 | 1 | 8 | 5 | 10 | 3 |
| FLDAS | 6 | 4 | 2 | 1 | 8 | 3 |
| GLEAM33a | 5 | 3 | 3 | 2 | 8 | 3 |
| TerraClimate | 1 | 1 | 4 | 3 | 5 | 2 |
| GLEAM33b | 0 | 0 | 3 | 2 | 3 | 1 |










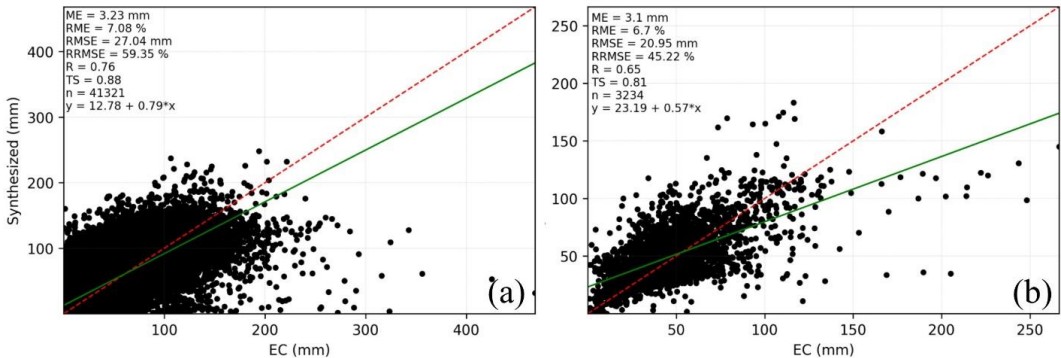

**Figure 14.** Monthly (**a**) and annually (**b**) synthesized ET against flux EC ET aggregated for all sites.

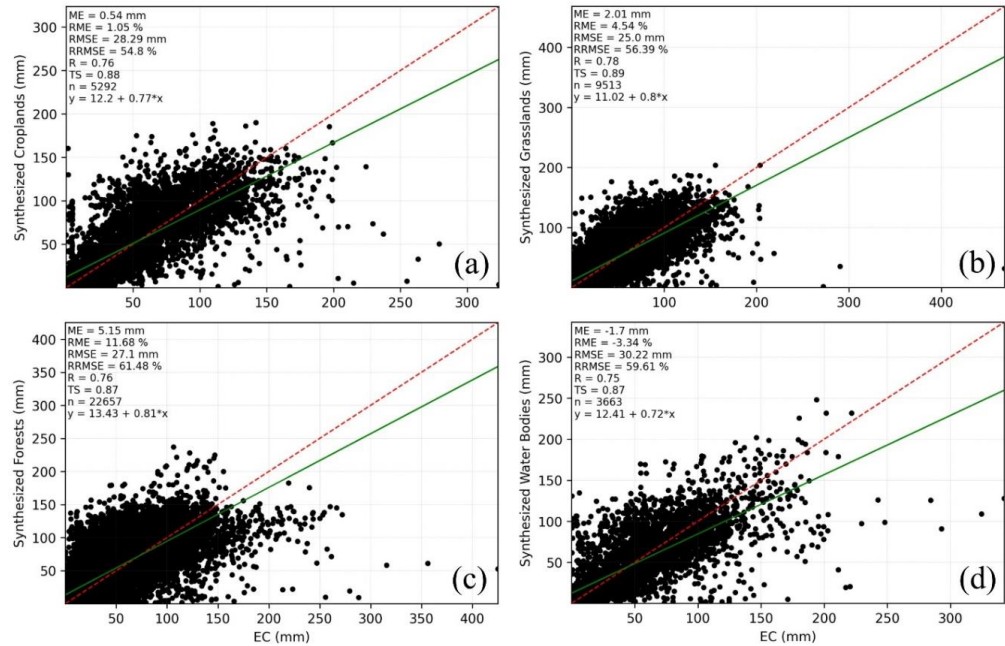

**Figure 15.** Monthly synthesized ET against flux EC ET aggregated for all sites for each land cover type (croplands: (**a**);
grasslands: (**b**); forest: (**c**); water bodies: (**d**)).

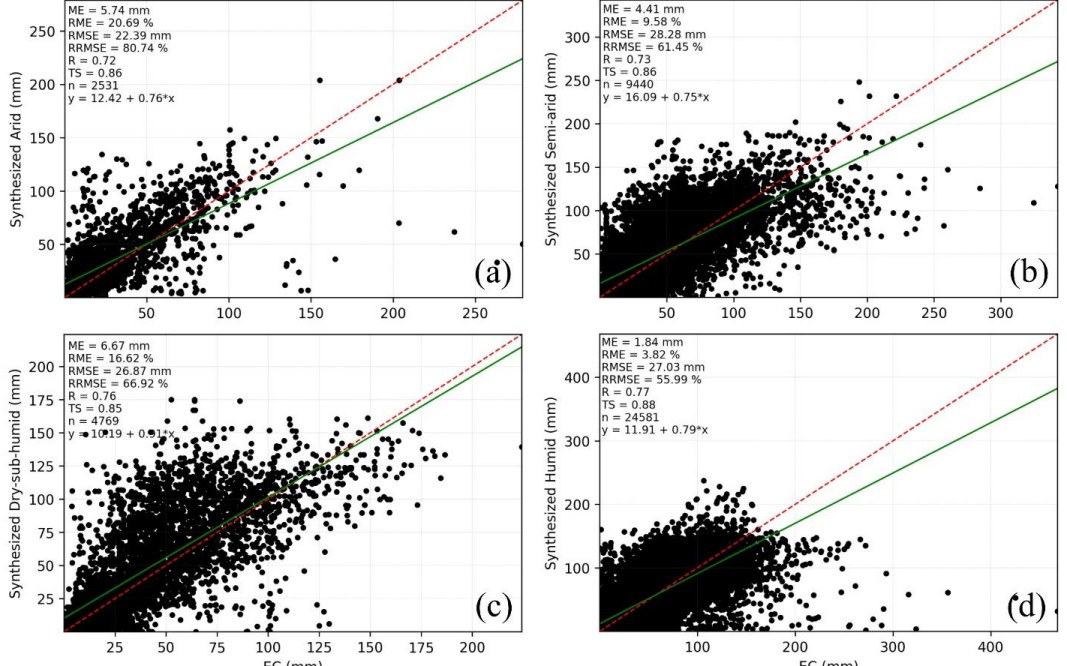

**Figure 16.** Monthly synthesized ET against flux EC ET aggregated for all sites for each climate class (arid: (**a**); semiarid: (**b**);
dry-sub-humid: (**c**); humid: (**d**)).

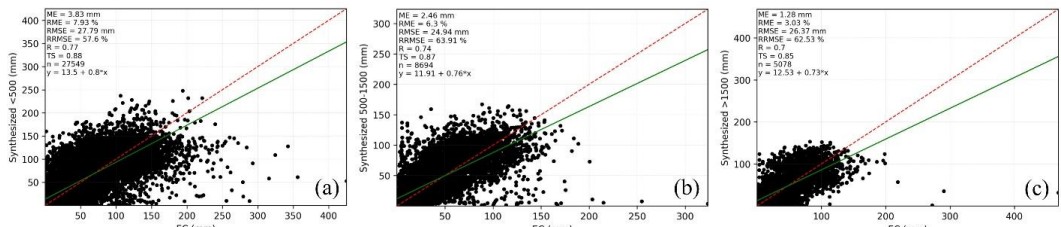

**Figure 17.** Monthly synthesized ET against flux EC ET aggregated for all sites for each elevation level (<500 m: (**a**); 500 m –
1500 m: (**b**); >1500 m: (**d**)).
Figure 18 presents a monthly comparison between the synthesized ET with the country-based ET products
over China and the United States as well as over the African continent. In general, the synthesized ET returned higher
agreement (R and TS) and accuracy (RMSE) with the flux EC ET than did the other ET products (China Terrestrial,
SSEBop, and FAO WaPOR). Moreover, it has lower biases over the United States and the African continent.

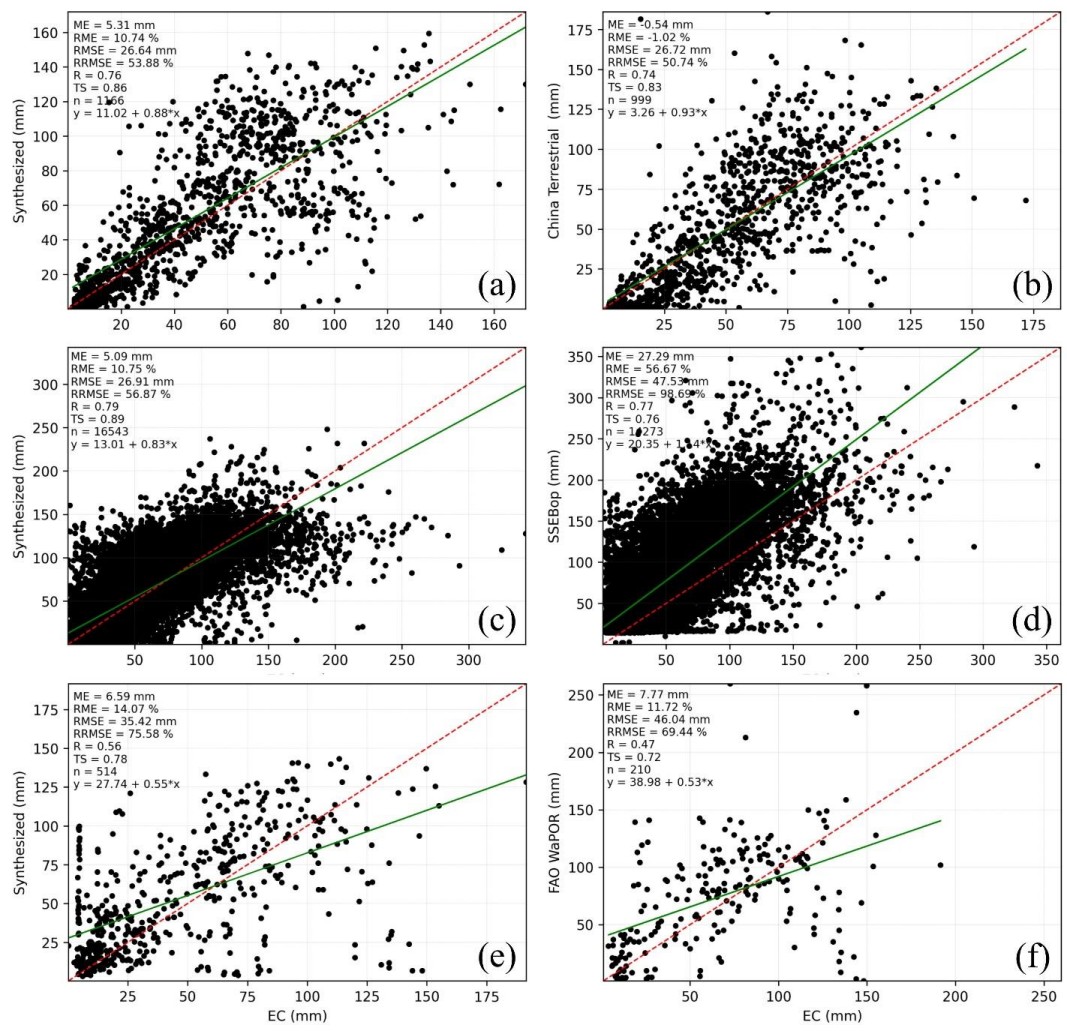

**Figure 18.** Monthly comparison between the synthesized ET (**a**, **c** and **e**) and China Terrestrial (**b**), SSEBop (**d**), and FAO WaPOR (**f**) products against flux EC ET aggregated for all sites over China (**a** and **b**), the USA (**c** and **d**) and the African continent (**e** and **f**).

## 5. Discussion

Since global land ET plays a paramount role in the hydrological cycle, its accurate estimation is essential for further studies. Although there are many global ET products that have been derived from remote sensing models, land surface models, and hydrological models, they differ in their algorithms, parameterization, and temporal span, and none of these products can be used for a long time with a reasonable spatial resolution and lower uncertainty. In this study, we ensemble the best-performing, currently available global ET products at a reasonable spatial resolution (kilometer) as one consistent global ET dataset covering a long temporal period. Users can use this dataset assuredly without looking at other datasets and performing additional assessments.





We used a high-quality dataset of global flux towers as a site-pixel-level validation for certain global ET
products (Leuning et al., 2008;Zhang et al., 2010;Ershadi et al., 2014;Michel et al., 2016) to assess them and select
the best products to create a synthesized ET covering a long temporal period. For that, a matrix of six validation
criteria and 26 comparison levels was created, and then levels one and two of the validation metrics were used to
select the best-performing products. Finally, by the simple mean of the products that performed best over the different
periods, the synthesized ET was created.
Among all global ET products investigated in this study, the products that performed best are PML,
GLDAS20, SSEBop, MOD16A2105, GLDAS21, SEBS, and NTSG (Table 6). From the perspective of all comparison
levels, the performance of these products varied, and no single product performed well across all land surface types
and conditions (Vinukollu et al., 2011a;Li et al., 2018). The PML represents the ET product with the highest
agreement, with lower ME (RME) and RMSE (RRMSE) values, followed by the synthesized ET (Tables 12 and 13);
however, it should be noted that PML estimates span a 15-yr period, while the synthesized ET presents longer
estimates from 1982 to 2019 (38 years).
The main advantage of the new dataset is that, for the first time, a synthesized remotely sensed ET product
with a reasonable spatial resolution and lower long-term uncertainties has been provided, where the maximum absolute
ME (RME) and RMSE (RRMSE) values are 13.94 mm (17.13 %) and 38.61 mm (47.45 %), respectively. Furthermore,
it agreed well ($R^2 > 0.70$) in 62 % of all comparison levels (Table 14). This dataset can provide ensemble ET estimates
for all land cover types, where MOD16A2105 do not provide ET estimates over water bodies and desert areas other
products are. Moreover, a comparison among the synthesized ET against China Terrestrial, SSEBop, and FAO
WaPOR ET products over China, the United States, and the African continent proved that the synthesized ET
outperformed these products in terms of a higher agreement, higher accuracies and lower biases. Hence, the
synthesized ET can play an essential role, especially for regional and global scale studies, over a long time (1892-
571  2019).

**Table 14.** Percentage of $R^2$ more than 0.70 and the maximum absolute value of ME (mm), RME (%) RMSE (mm), and RRMSE
(%) across all comparisons levels (01-26) of the highly preformed ET products and the synthesized ET.

| Dataset | $R^2$>0.7 (%) | ME | RME | RMSE | RRMSE |
|---|---|---|---|---|---|
| PML | 65 | 7.64 | 12.22 | 36.28 | 44.30 |
| Synthesized | 62 | 13.94 | 17.13 | 38.61 | 47.45 |
| GLDAS20 | 42 | 9.73 | 23.02 | 39.53 | 49.32 |
| SSEBop | 42 | 21.82 | 26.07 | 48.14 | 57.50 |
| MOD16A2105 | 42 | 12.89 | 51.06 | 42.78 | 53.27 |
| GLDAS21 | 35 | 13.69 | 22.07 | 47.84 | 58.32 |
| NTSG | 23 | 14.46 | 86.35 | 40.50 | 50.26 |


The synthesized ET used SSEBop ET for the years 2018 and 2019 and NTSG from 1982 to 2000 because
NTSG is the only remotely sensed global ET product available and has a good spatial resolution compared to
GLDAS20. It is the simple mean of MOD16A2105 and NTSG for the years 2001 and 2002 and the simple mean of
PML and SSEBop between 2003 and 2017 (see Tables 7 and 9).
Because the ET was synthesized during the first and second decades as well as the year 2000 based on
resampled NTSG to a 1 km spatial resolution to be comparable with other products, future improvements may be
focused on downscaling NTSG during this period to enhance the product proposed in this paper.
**6. Data availability**
All data used in this study are freely available; see Sect. 2 and Appendix A. The synthesized ET is available
in https://doi.org/10.7910/DVN/ZGOUED (Elnashar et al., 2020) and as GEE application from the following link:
https://elnashar.users.earthengine.app/view/synthesizedet also, it can be accessed  in the GEE JavaScript editor (the
updated link embedded in the GEE application interface). Through this application, the user can query and display as
well as download the synthesized ET. It should be noted that SSEBop and NTSG datasets are not available in Earth
Engine so they were uploaded as assets in GEE for this purpose.
**7. Conclusion**
In the current study, a site-pixel-level validation was conducted for certain global ET products across a variety
of land surface types and conditions to select the best performing ET products and then produce a global long-term
synthesized ET dataset. To apply a comprehensive evaluation from different perspectives, land cover types, climate
and elevations were classified into five, four, and three classes, respectively. According to six comprehensive
validation criteria, the evaluated ET products ranked based on the lowest error metrics and highest accuracy and
consistency over different classification levels to choose the ensemble members over different times.
Concerning the study investigation, PML, GLDAS20, SSEBop, MOD16A2105, GLDAS21, SEBS, and
NTSG were ET products that performed best. Although no product performed best in terms of all selected validation
criteria in all classification levels, the synthesized ET produced from PML, SSEBop, MOD16A2105 and NTSG had
high agreements and accuracies with low biases over most of the land surface types and conditions. In addition, this
study provides ET estimates from 1982 to 2019 and for all land cover types. Furthermore, it performed well when
compared with country-based and continental ET products over China, the United States and the African continent.
The results from this study provide a better understanding of the high performing ET products in each land
cover type, elevation level and climate region as well as at monthly, annual and interannual time steps. Hence, this
study provides an ET product that can be used to improve the quality of ET at the global level and, consequently, can
be used to improve agriculture, water resource management, and climate change studies.
**Author Contribution:** Abdelrazek Elnashar was responsible for experimental designing, manuscript preparation, and
data processing and presentation. Linjiang Wang, Dr. Weiwei Zhu, and Dr. Hongwei Zeng contributed to data
processing. Prof. Dr. Bingfang Wu contributed to conceptual designing, reviewing of the manuscript, funding
acquisition, and project administration.
**Acknowledgments**: This research was financially supported by the National Key Research and Development Program
of China (Grant No. 2016YFA0600303), the National Natural Scientific Foundations of China (grant numbers:
41991232) and the Key Research Program of Frontier Sciences, CAS (grant numbers: QYZDY-SSW-DQC014).



**Conflicts of Interest**: The authors declare that they have no conflict of interest.
**Appendix A**
A summary of ET datasets used in this research is presented here. It should be noted that except for SSEBop,
SEBS, NTSG ET, and GLEAM, which downloaded from their providers, other datasets are available in Earth Engine
Data Catalog through the following link https://developers.google.com/earth-engine/datasets/catalog. Each dataset in
GEE has Earth Engine Snippet as following:
MOD16A2 ET V6: ee.ImageCollection("MODIS/006/MOD16A2")
MOD16A2 ET V105: ee.ImageCollection("MODIS/NTSG/MOD16A2/105")
PML ET: ee.ImageCollection("CAS/IGSNRR/PML/V2")
GLDAS ET V20: ee.ImageCollection("NASA/GLDAS/V20/NOAH/G025/T3H")
GLDAS ET V021: ee.ImageCollection("NASA/GLDAS/V021/NOAH/G025/T3H")
FLADS ET: ee.ImageCollection("NASA/FLDAS/NOAH01/C/GL/M/V001")
TerraClimate ET: ee.ImageCollection("IDAHO_EPSCOR/TERRACLIMATE")
**MOD16 ET**
The Moderate Resolution Imaging Spectroradiometer (MODIS) Global Evapotranspiration Project
(MOD16A2) estimates terrestrial ET as the sum of evaporation and plant transpiration. MOD16A2 ET uses the
Penman-Monteith model, which includes MODIS remotely sensed data (e.g., vegetation, surface albedo, and land
cover classification) and daily meteorological reanalysis. There are two products of MOD16A2 ET (V6 and V105)
with an 8-day temporal resolution, but they differ in their spatial resolution and temporal coverage (Mu et al., 2011;Mu
et al., 2014a). V6 spans from 2001 until now with a 500 m × 500 m spatial resolution and is provided by NASA LP
DAAC at the USGS EROS Center; it can be downloaded from https://doi.org/10.5067/MODIS/MOD16A2.006. V105
estimates span the period from 2001 to 2014 with a 1000 m × 1000 m spatial resolution and are provided by the
Numerical Terradynamic Simulation Group (NTSG) at the University of Montana in conjunction with the NASA
Earth Observing System (Mu et al., 2014b).
**PML ET**
**The** Penman-Monteith Leuning (PML) ET product partitions ET into three components: plant transpiration,
soil evaporation and intercepted rainfall by the canopy as well as water evaporation. PML data span from 2002 to
2017 with a 500 m × 500 m spatial resolution and an 8-day temporal resolution (Zhang et al., 2019).
**SSEBop**
The operational Simplified Surface Energy Balance (SSEBop) model is based on the Simplified Surface
Energy Balance (SSEB) approach with a unique parameterization for operational applications. Using a thermal index
approach, it combines ET fractions generated from remotely sensed MODIS land surface temperature, acquired every



10 days, with reference ET from global weather datasets. The SSEBop uses predefined, seasonally dynamic, boundary
conditions that are unique to each pixel for the hot and cold reference points (Senay et al., 2007;Senay et al.,
2011;Senay et al., 2013;Senay et al., 2020). SSEBop estimates are from 2003 with a 0.0096°×0.0096° (≈1 km) spatial
resolution and a monthly temporal resolution. Data were provided by The Early Warning and Environmental
Monitoring Program via the United States Geological Survey and can be downloaded from the following website
https://earlywarning.usgs.gov.
**SEBS**
The Surface Energy Balance System (SEBS) is an approach designed to estimate ET from the evaporative
fraction using satellite remote sensing augmented with meteorological data at corresponding scales (Su, 2002).
MODIS-LST and the Normalized Difference Vegetation Index (NDVI), GLASS-LAI, GLAS global forest height,
GlobAlbedo, and ERA-Interim meteorological data have been used in these ET calculations with the revised SEBS
algorithm (Chen et al., 2013;Chen et al., 2014a;Chen et al., 2019). SEBS is available during the period from 2000 to
2017 with a 5 km × 5 km spatial resolution and monthly temporal resolution. It is copyrighted by the Institute of
Tibetan Plateau Research, Chinese Academy of Sciences and is available at http://en.tpedatabase.cn.
**NTSG ET**
The Numerical Terradynamic Simulation Group (NTSG) ET data are based on an algorithm that estimates
transpiration from the canopy and evaporation from soil using a modified Penman-Monteith model and evaporation
from open water using a Priestley-Taylor model. These algorithms were applied globally using the Advanced Very
High-Resolution Radiometer (AVHRR) Global Inventory Modeling and Mapping Studies (GIMMS) NDVI,
NCEP/NCAR Reanalysis daily surface meteorology, and NASA/GEWEX Surface Radiation Budget Release-3.0 solar
radiation inputs (Zhang et al., 2009;Zhang et al., 2010). NTSG estimates cover a period from 1982 to 2013 at a spatial
resolution of 8 km × 8 km and a monthly temporal resolution. It is produced by NTSG at the University of Montana
and can be retrieved from http://files.ntsg.umt.edu/.
**GLEAM**
The Global Land Evaporation Amsterdam Model (GLEAM) is physically based on algorithms that estimate
ET components separately (i.e., transpiration, interception loss, bare soil evaporation, snow sublimation, and open-
water evaporation). The potential evaporation is estimated by the Priestley and Taylor equation based on observations
of surface net radiation and near-surface air temperature and is then converted into actual evaporation based on the
evaporative (soil) stress factor. The soil stress factor is based on microwave vegetation optical depth and simulated
root-zone soil moisture calculated from a multilayer water balance model. Separately, interception loss is calculated
based on vegetation and rainfall observations. There are two datasets available for GLEAM (i.e., v3.3a, and v3.3b)
that differ only in their forcing and temporal coverage. v3.3a spans from 1980 to 2018 and relies on reanalysis radiation
and air temperature, a combination of gauge-based, reanalysis and satellite-based precipitation, and satellite-based
vegetation optical depth, while v3.3b spans from 2003 to 2018, and its forcing factors are the same as v3.3a except



for radiation and air temperature, which are based on remotely sensed data. Both v3.3a and v3.3b estimates are
provided at a monthly temporal resolution and a 0.25°×0.25° (≈25 km) spatial resolution (Miralles et al.,
2011b;Miralles et al., 2011a;Martens et al., 2017).

**GLDAS ET**

The Global Land Data Assimilation System (GLDAS) generates optimal fields of land surface states and
fluxes using advanced land surface modeling and data assimilation techniques by ingesting satellite and ground-based
observational data products. GLDAS Version 2 has two components (GLDAS-2.0 and GLDAS-2.1) with a
0.25°×0.25° (≈25 km) spatial resolution and a 3-hr temporal resolution. GLDAS-2.0 is reprocessed with the updated
Princeton Global Meteorological Forcing Dataset and upgraded Land Information System Version 7. The model
simulation was initialized from January 1, 1948, to December 31, 2010, using soil moisture and other state fields from
the LSM climatology for that day of the year. The simulation used the common GLDAS datasets for land cover
(MCD12Q1), land-water mask (MOD44W), and soil texture and elevation (GTOPO30). The GLDAS-2.1 simulation
started on January 1, 2000, and lasted until December 31, 2019, using the conditions from the GLDAS-2.0 simulation.
This simulation was forced with the National Oceanic and Atmospheric Administration (NOAA)/Global Data
Assimilation System (GDAS) atmospheric analysis, disaggregated Global Precipitation Climatology Project (GPCP)
precipitation, and Air Force Weather Agency's AGRicultural METeorological modeling system (AGRMET) radiation.
The MODIS-based land surface parameters were used in the current GLDAS-2.x products, while the AVHRR base
parameters were used in previous GLDAS-2 products before October 2012 (Rodell et al., 2004).

**FLDAS ET**

The Famine Early Warning Systems Network (FEWS NET) Land Data Assimilation System (FLDAS)
dataset uses remotely sensed and reanalysis inputs to drive land surface models. It includes information on many
climate-related variables, including evapotranspiration, moisture content, humidity, average soil temperature, and total
precipitation rate. For forcing data, this FLDAS dataset uses a combination of the new version of Modern-Era
Retrospective analysis for Research and Applications version 2 (MERRA-2) data and Climate Hazards Group
InfraRed Precipitation with Station data (CHIRPS), a quasi-global rainfall dataset designed for seasonal drought
monitoring and trend analysis (McNally et al., 2017). FLDAS is provided at a 0.1°×0.1° (≈10 km) spatial resolution
and monthly temporal resolution during the period 1982-2019.

**TerraClimate ET**

TerraClimate ET is estimated based on monthly one-dimensional soil water balance for global terrestrial
surfaces, which incorporates evapotranspiration, precipitation, temperature, and interpolated plant extractable soil
water capacity. The water balance model is very simple and does not account for heterogeneity in vegetation types or
their physiological responses to changing environmental conditions (Abatzoglou et al., 2018). TerraClimate estimates
are provided at a monthly temporal resolution from 1958 to 2018 and 0.041°×0.041° (≈5 km) grid cells.





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
