# Peer review of "Synthesis of Global Actual Evapotranspiration from 1982 to 2019"

_Earth System Science Data, 2020_

## Referee Comment (RC1) · Anonymous Referee #1 · 4 Aug 2020

This manuscript is very interesting and valuable in developing a global accurate ET dataset. Currently, there are many global or regional ET datasets, but their performances vary across different regions. This manuscript provides an insightful approach in processing these datasets ensemble. However, there are many procedures to be clarified to inform the readers. Major comments: 1)I don't understand the meanings of "best first and second levels" and "levels one and two validation metrics". These two phrases have appeared many times and is vital important to understand the synthesis procedure. If I understand this correctly, the performance metrics in Tables 5-8 were used to select two or three best ET datasets and the new dataset is produced by averaging these two or three datasets. A figure of processing procedure may be helping. 2)The second major problem is the validation data. By reading this manuscript,

it could be confirmed that the observed EC ET data serve as validation data in evaluating and ranking the 12 ET datasets and also the validation data in evaluating the proposed Global Actual Evapotranspiration dataset. There could be overfitting effect. It is like we using the same dataset as calibration data and validation data at the same time. Therefore, cross-checking method should be applied. For example, 2/3 of the EC sites be used to evaluate the ET datasets and 1/3 of EC sites to validate. 3)Another question that should be discussed is the scale problem. The EC sites normally work on a very limited area and can only present the ET condition of a small region. The related uncertainty should be discussed in the manuscript. Some minor comments. 1)Line 11: What do you mean by "they produce different levels of uncertainties" 2)In the abstract, the synthesization method should be indicated clearly. 3)Line 74: check the time period 4)Line 258: the title of subplot c 5)Line 371-392: Different datasets were selected due to data availability. That means for each period, for example before 2003 and 2003-2017, different datasets were used. My concern is that the ensemble means/variations may differ greatly. An adjustment in the period mean/variation should be considered. 6)Some table and figure captions are similar. For example, Table 5 and Table 12. The major differences between them are the time period, which should be clearly indicated.

---

## Author Comment (AC1) · 24 Aug 2020

Response to Referee 1's Comments.

**************** General comments ****************

This manuscript is very interesting and valuable in developing a global accurate ET dataset. Currently, there are many global or regional ET datasets, but their performances vary across different regions. This manuscript provides an insightful approach in processing these datasets ensemble. However, there are many procedures to be clarified to inform the readers.

»Answer: Thank you very much for your positive comments and suggestions which for sure significantly improved the manuscript.

**************** Major comments ****************

I don't understand the meanings of "best first and second levels" and "levels one and two validation metrics". These two phrases have appeared many times and are vitally important to understand the synthesis procedure. If I understand this correctly, the performance metrics in Tables 5-8 were used to select two or three best ET datasets and the new dataset is produced by averaging these two or three datasets. A figure of processing procedure may be helping.

»Answer: Thank you for pointing this out. The level one of validation metrics has the highest R and TS values and the lowest ME, RME, RMSE, and RRMSE while the level two of validation metrics has the highest R and TS values and the lowest ME, RME, RMSE, and RRMSE after level one. You are right, the performance metrics in Tables 5-8 were used to select two or three best ET datasets and the new dataset is produced by averaging these two or three datasets. For that, Figure 1, that shown below, was created to preview the synthesization method and is included in the revised manuscript under Section 3.2. Hence, we rewrote Section 3.2, as follows:

"There are 6 validation metrics including R, TS, ME, RME, RMSE, and RRMSE. The validation values of 6 metrics are categorized into levels. The level one of validation metrics has the highest R and TS values and the lowest ME, RME, RMSE, and RRMSE while the level two of validation metrics has the highest R and TS values and the lowest ME, RME, RMSE, and RRMSE after level one. For that, R and TS sorted descending while ME, RME, RMSE, and RRMSE sorted ascending (Fig.1a) then the corresponding ET product of each validation metric saved in a new table to be used to fill in Fig.1b. The current study proposes three steps to develop a synthesized global ET dataset. First, the ET datasets are compared based on 6 validated metrics to generate a matrix to indicate level one and two of the validation metrics of all ET products over all comparison levels (Fig.1b). For each level, there are 6 validation metrics in rows and 26 ET values of different time periods and underlying conditions in columns (comparison levels), including monthly average (01), annual average (02), monthly (January-

December: 03-14), land cover types (15-19), climate classes (20-23), and elevation levels (24-26). Thus, the total number of cells is 156 for each level. Each cell in the matrix represents one of twelve ET products that belong to this level. Then, to select ET data for further synthesis, the number and percentage of ET product occurrence at matrix (Fig.2b) of level one and two were calculated (Fig.1c). ET products were ranked in descending order based on the occurrence percentage of levels one and two (the last column in Fig.1c). Finally, the first two or three highly ranked ET products were selected to incorporate into the ensemble ET. For that, the selected ET products were resampled to a comparable spatial resolution if needed, and the average was used as the synthesized ET value."

The second major problem is the validation data. By reading this manuscript, it could be confirmed that the observed EC ET data serve as validation data in evaluating and ranking the 12 ET datasets and also the validation data in evaluating the proposed Global Actual Evapotranspiration dataset. There could be an overfitting effect. It is like we use the same dataset as calibration data and validation data at the same time. Therefore, a cross-checking method should be applied. For example, 2/3 of the EC sites be used to evaluate the ET datasets and 1/3 of EC sites to validate.

»Answer: Thank you very much for your comment. You are right, it needs to split the in-situ data into 2 groups for calibration and validation. However, we do not calibrate ET products. We use in-situ data to see which one is performing better than others. Once ET products are selected, then we synthesize them into one and use in-situ data to validate to see if the synthesized data is better. Furthermore, From Tables 1 and 3, the flux EC ET sites, as well as the 12 ET products, are available in different periods. For evaluating each ET product the matched periods between EC sites and ET datasets were used (Xu et al. 2019; Li et al. 2018), that is why RME and RRMSE are included in the validation metrics. Further, the synthesized ET represented by the mean of PML and SSEBop about 60% (2003 to 2017) and the mean of NTSG and MOD16A2105 about 8% (2002-2002) indicating 68% of the synthesized ET are new data. For that,

we used the matched period's method aiming to validate the new product under the same conditions of the experiment. We agreed with this method because we did not incorporate the flux EC ET data into the synthesized ET, it just serves as a ruler to prove to what extend each ET product works well in all comparison levels. Moreover, we used three regional ET datasets for comparison of consistent agreement over China, the United States, and the African continent to ensure the proposed product works well.

Another question that should be discussed is the scale problem. The EC sites normally work in a very limited area and can only present the ET condition of a small region. The related uncertainty should be discussed in the manuscript.

»Answer: Thank you for your very thoughtful comment. This is a common issue. The best way to validate the ET datasets is to use closure watershed water balances, however, these data set are quite a few. Flux EC ET data has its footprint, covering a larger area, but hard to match with a pixel. This issue still needs fundamental study. For that, we added Lines 61-64, as follows:

"Although flux EC ET is commonly flawed, particularly concerning energy balance closure at some sites (Foken, 2008; Helgason and Pomeroy, 2012), relatively short periods, and sparse spatial coverage, it is the most direct method for measuring the exchange between the surface and the atmosphere in different ecosystems (Foken et al., 2012; Baldocchi, 2014). Thus, site-pixel-level validation of certain ET products against flux EC ET as typically observed data has been performed by several studies in specific regions......"

**************** Minor comments ****************

Line 11: What do you mean by "they produce different levels of uncertainties?"

»Answer: Thank you for that comment. We rewrote the sentence (Lines 9-11) to be clearer, as follows:

"Although it is difficult to estimate ET over a large scale and for a long time, there are

several global ET datasets available with uncertainty associated with various assumptions regarding their algorithms, parameters, and inputs".

In the abstract, the synthesization method should be indicated clearly.

»Answer: Thank you for your cogent advice. We agree and have indeed done that (Lines 12-15), as follows:

"Through a site-pixel evaluation of 12 global ET products over different time periods, land surface types, and conditions, the high performing products were selected for synthesis of the new dataset using a high-quality flux eddy covariance covering the entire globe."

Line 74: check the time period.

»Answer: Thank you for pointing this out. We changed 1998-1995 to 1989-1995.

Line 258: the title of subplot c.

»Answer: Thank you for pointing this out. We changed RMSE (mm): (d) to RMSE (mm): (c).

Line 371-392: Different datasets were selected due to data availability. That means for each period, for example before 2003 and 2003-2017, different datasets were used. My concern is that the ensemble means/variations may differ greatly. An adjustment in the period mean/variation should be considered.

»Answer: Thank you for your very thoughtful comment. Although we agree with you, this time series adjustment is very important and should be done in the future. Therefore, we have added Lines 572-574, as follows:

"since different datasets were selected due to data availability, also future improvements may be focused on the adjustment of the ensemble means particularly for long-term pixel-based studies."

Some tables and figure captions are similar. For example, Table 5 and Table 12. The major differences between them are the time period, which should be clearly indicated.

»Answer: We appreciate your advice. Tables and figures caption has been revised (Figures 6 and 13; Tables 6-9, 12, 13).

12 global ET datasets  645 flux EC ET sites

6 validation metrics
(ME, RME, RMSE, RRMSE, R and TS)

[Figure]

(a)

| Sorting | Ascending | Ascending | Ascending | Ascending | Descending | Descending |
|---|---|---|---|---|---|---|
| Validation metrics of level 1 | ME1 | RME1 | RMSE1 | RRMSE1 | R1 | TS1 |
| Validation metrics of level 2 | ME2 | RME2 | RMSE2 | RRMSE2 | R2 | TS2 |
|  |  |  |  |  |  |  |
| Validation metrics of level 12 | ME12 | RME12 | RMSE12 | RRMSE12 | R12 | TS12 |

ME1 < ME2
RME1 < RME2
RMSE1 < RMSE2
RRMSE1 < RRMSE2
R1 > R2
TS1 > TS2

(b)

| Levels | Metrics | 01 | 02 |  | 26 |
|---|---|---|---|---|---|
|  | ME |  |  |  |  |
|  | RME |  |  |  |  |
| Level 1 metrics | RMSE |  |  |  |  |
|  | RRMSE |  |  |  |  |
|  | R |  |  |  |  |
|  | TS |  |  |  |  |
|  | ME |  |  |  |  |
|  | RME |  |  |  |  |
| Level 2 metrics | RMSE |  |  |  |  |
|  | RRMSE |  |  |  |  |
|  | R |  |  |  |  |
|  | TS |  |  |  |  |

→ Comparison levels (from 01 to 26 )

→ Corresponding ET product which presents the minimum ME for all sites-months, cell color is matched with this product cell color in table (c).

Synthesized ET is the mean of top first 2 or 3 datasets

(c)

| ET products | Occurrence in level 1 | | Occurrence in level 2 | | Total | |
|---|---|---|---|---|---|---|
|  | count | % | count | % | count | % |
| Sorting | no | no | no | no | no | Descending |
| Product 1 | a | (a /6*26) *100 | b | (b/6*26)*100 | c= a+b | (c/6*26*2)*100 |
| Product 2 |  |  |  |  |  |  |
|  |  |  |  |  |  |  |
| Product 12 |  |  |  |  |  |  |

**Fig. 1.** Flowchart of the synthesization method

---

## Referee Comment (RC2) · Rafat Ali (Referee) · 22 Dec 2020

The present article proposes a long-term synthesized ET product at a kilometer spatial resolution and monthly temporal resolution from 1982 to 2019. The authors made trial application of GIS and remotely sensed data to reach the proposed aim of their study. The presented article would be a good piece of work by supporting the conclusion with the obtained findings.

---

## Author Comment (AC2) · 31 Dec 2020

Response to Referee 2's Comments

The present article proposes a long-term synthesized ET product at a kilometer spatial resolution and monthly temporal resolution from 1982 to 2019. The authors made a trial application of GIS and remotely sensed data to reach the proposed aim of their study.

» Answer: Thank you very much for your positive comments and suggestions which for sure significantly improved the manuscript.

The presented article would be a good piece of work by supporting the conclusion with the obtained findings.

» Answer: Thank you for your very thoughtful comment. We have added the obtained findings to the Conclusions section as follows: "The average annual ET from 1982–2019 is 567 mm year–1. Although no product performed better in terms of all selected validation criteria in all classification levels, PML, GLDAS20, SSEBop, MOD16A2105, GLDAS21, SEBS, and NTSG are the sequence of their performances. The synthesized ET from PML, SSEBop, MOD16A2105 and NTSG agreed with the flux EC ET with R-values higher than 0.70, a maximum ME (RME) of 13.94 mm (17.13%) and a maximum RMSE (RRMSE) of 38.61 mm (47.45%) over 62% of all comparisons levels, as remotely sensed based ET product spanning from 1982 to 2019 with highest agreements, accuracies and lower biases over most of the land surface types and conditions. It performs well when compared with country-based and continental ET products over China, the United States and the African continent. However, the further synthesis of local ET products is encouraged if regional ET products are available.".